# Infinitesimal sulfur fusion yields quasi-metallic bulk silicon for stable and fast energy storage

Jaegeon Ryu[1,5], Ji Hui Seo[2,5], Gyujin Song [2], Keunsu Choi[2], Dongki Hong[2], Chongmin Wang [3], Hosik Lee[4], Jun Hee Lee[2] & Soojin Park[1]

A fast-charging battery that supplies maximum energy is a key element for vehicle electrification. High-capacity silicon anodes offer a viable alternative to carbonaceous materials, but they are vulnerable to fracture due to large volumetric changes during charge–discharge cycles. The low ionic and electronic transport across the silicon particles limits the charging rate of batteries. Here, as a three-in-one solution for the above issues, we show that small amounts of sulfur doping (<1 at%) render quasi-metallic silicon microparticles by substitutional doping and increase lithium ion conductivity through the flexible and robust self-supporting channels as demonstrated by microscopy observation and theoretical calculations. Such unusual doping characters are enabled by the simultaneous bottom-up assembly of dopants and silicon at the seed level in molten salts medium. This sulfur-doped silicon anode shows highly stable battery cycling at a fast-charging rate with a high energy density beyond those of a commercial standard anode.

[1] Department of Chemistry, Division of Advanced Materials Science, Pohang University of Science and Technology (POSTECH), Pohang 37673, Republic of Korea. [2] Department of Energy Engineering, School of Energy and Chemical Engineering, Ulsan National Institute of Science and Technology (UNIST), Ulsan 44919, Republic of Korea. [3] Environmental Molecular Sciences Laboratory, Pacific Northwest National Laboratory, 902 Battelle Boulevard, Richland, WA 99354, USA. [4] Department of Mechanical Engineering, School of Mechanical and Nuclear Engineering, Ulsan National Institute of Science and Technology (UNIST), Ulsan 44919, Republic of Korea. [5] These authors contributed equally: Jaegeon Ryu, Ji Hui Seo. Correspondence and requests for materials should be addressed to H.L. (email: hslee@unist.ac.kr) or to J.H.L. (email: junhee@unist.ac.kr) or to S.P. (email: soojin.park@postech.ac.kr)

When silicon (Si) is heavily doped with chalcogen family elements (e.g., S, Se, and Te) at a concentration exceeding the equilibrium solid solubility, it experiences the insulator-to-metal transition (IMT); thus, it shows great potential for optoelectronic applications such as infrared detection and intermediate-band solar cells[1,2]. At present, such a supersaturated structure has been exclusively realized by an intricate combination of ion implantation, pulsed-laser-induced melting, and rapid solidification to activate the dopants and restore the lattice damaged by accelerated ions[3]. However, this series of step requires high-cost facilities, provides shallow doping depth of <1 μm, and poses serious hazards, although it is currently introduced to process in semiconductor devices.

Moreover, refining a commercial grade Si anode is of great technological importance to extend the energy limit of lithium (Li)-ion batteries with a high specific capacity (> 3500 mA h g$^{-1}$) and a low operation potential (< 0.4 V vs Li/Li$^+$) (refs. [4,5]). However, the severe expansion of Si is inevitable upon Li insertion that poses a major challenge. Nevertheless, at the single-specimen scale, this bottleneck has been resolved by nanoscale designs incorporated with sufficient porosity[6–12], pre-reserved void spaces[13–17] and composite formation with conductive carbon buffers[18–23], which prevent mechanical fracture[24,25] and extend battery life[26,27]. Despite such breakthroughs, poor initial Coulombic efficiency (ICE) causes a significant energy loss. In addition, materials with a low tap density are unfavorable to achieve high-energy-density batteries.

Consequently, a bulk Si microparticle (SiMP) anode has been directly utilized for its low cost, practical availability and higher ICE in a way of either coalescence or confinement methods by introducing the multifunctional binders or external coating layers[28–32]. However, building a structure for SiMP anodes that is durable beyond the nanoscale compartment is a considerable obstacle facing the battery community. Even when the anode is outfitted with an anti-pulverization structure, Li-ion diffusion through such a large domain as well as insufficient electronic conduction limits further use of SiMP anodes in forthcoming applications.

In this study, we report a low-temperature sulfur fusion approach to a quasi-metallic Si (QMS) anode with a large average particle size of 3 μm and a hollow spherical structure with controllable doping levels of sulfur. Unlike previous approaches based on the forced insertion of dopants, this spontaneous co-growth pathway of the reduced silicon and sulfur seeds from the low-temperature reduction reactions in the molten salt medium contributes to a uniform doping environment by a small quantity of sulfur substitution into the Si crystal and interior channel formation buffered by flexible and robust sulfur chains. The substituted sulfur dopants significantly increase the electronic conductivity even featuring the metallic nature as the doping concentration increases, while the self-supporting channels originated from the sulfur chains provide a diffusion channel for lithium ions. The electronically and ionically conductive QMS shows high initial reversibility during the first charge–discharge cycles despite its bulk particle size. The electrochemically generated lithium sulfides help to retain the metallic property, thereby extending the cycling life of battery at a fast-charging rate with a high-energy density in both half and full-cell systems.

## Results
### Simultaneous seeds growth-enabled uniform sulfur doping.
In principle, aluminum chloride (AlCl$_3$) used here as a metal salt ($T_m$ of 192 °C) as well as molten salt medium can solvate bulk aluminum (Al) to form the highly reactive Al-AlCl$_3$ complex structure that spontaneously reduces different types of silica (e.g.,

cost-effective clay minerals and commercial bulk SiO$_2$) in two thermodynamically stable pathways. Activated AlCl* from the ligand of the complex adsorbs on the oxygen atom of these compounds and then generates unusual byproducts of aluminum oxychloride (AlOCl) along with the formation of Si seed as demonstrated previously[7]. The complex species also react with an additional metal salt (MgSO$_4$), selectively dissociating the oxygen atoms from the salt crystal structure and thereby yield isolated magnesium (Mg) and sulfur. The presence of metallic Mg, which acts as the metal center of the complex, leads to the evolution of secondary byproducts of MgAl$_2$Cl$_8$ as clearly evidenced by X-ray diffraction (XRD) analysis of crude products after the reduction reactions (Supplementary Fig. 1). Further, protuberant XRD peaks observed at near 20° and 40°, correspond to the amorphous sulfur clusters while still buried with sharp peaks for either metallic Al or Si, which are indicative of the simultaneous formation of Si and sulfur seeds. Strong reducing power of the complex completely disintegrate initial precursors into the active Si and sulfur at the atomic level as suggested above but still stabilized by and embedded in the fluidic molten salt medium. Afterward, abundant byproducts set the clustering environment for each atom to grow into the seeds that have relatively free motion in the medium and eventually they are assembled into the spherical structure to reduce the surface energy of particles as the reactor cools down to ambient temperature. During the recrystallization process, bonds between the two seed fractions and neighboring seeds spontaneously saturate, while concurrent sulfur anchoring on either side of the crystallized Si surface restrains the pore filling (Fig. 1a). The resulting self-supporting channels facilitate the fast diffusion of Li ions, and the directly substituted sulfur atoms in the Si crystal structure change the nature of Si into the quasi-metallic state. Without the sulfur fusion in the interatomic spaces, such defect sites completely merge (Fig. 1b), although both systems produce hollow and porous frameworks via localized Ostwald ripening process.

Unusual sulfur fusion enables the deep and uniform doping of sulfur dopants with different coordination states into the porous SiMP in a processible and cost-effective way (Fig. 1c–e and Supplementary Fig. 2). These 1–5 μm QMS particles have tunable sulfur concentrations between 0.1 and 0.7 at% with an infusion limit due to the excessive sulfur loss through silicon sulfide formation (Supplementary Fig. 3 and Supplementary Note 1). Even with the higher doping level than that allowed by the implantation process, our approach maintains the cubic crystalline phase of Si, in contrast with the polymorphic structure that evolves in hyperdoped Si wafers (Supplementary Figs. 4 and 5) (ref. [2]). In addition, electron energy loss spectroscopy (EELS) indicates that the QMS contains sulfur in various states other than elemental sulfur, and the Si L-edge of the QMS shifts upward from 99.4 to 99.9 eV, resulting from the sulfur that fuses into the Si in two separate manners (Fig. 1f). Interestingly, we observed wide defect clouds over the well-defined Si atomic structures that are assumed to be sulfur-fusion-induced channels (or defects) and lattice distortion owing to substitutional sulfur atoms (Fig. 1g, h). Two individual sulfur couplings in the QMS samples are separately investigated in detail in terms of their metallicity and ionic channels.

**Quasi-metallic transition in Si microparticles**. The increased electronic conductivity of the QMS was revealed by measuring its single-particle conductance during in situ probing and its bulk conductivity in pellet form (supplementary materials, methods). Samples were mounted on a tungsten electrode connected to another electrode to apply voltage sweeps, and their current responses were recorded. Representative current–voltage (I–V)

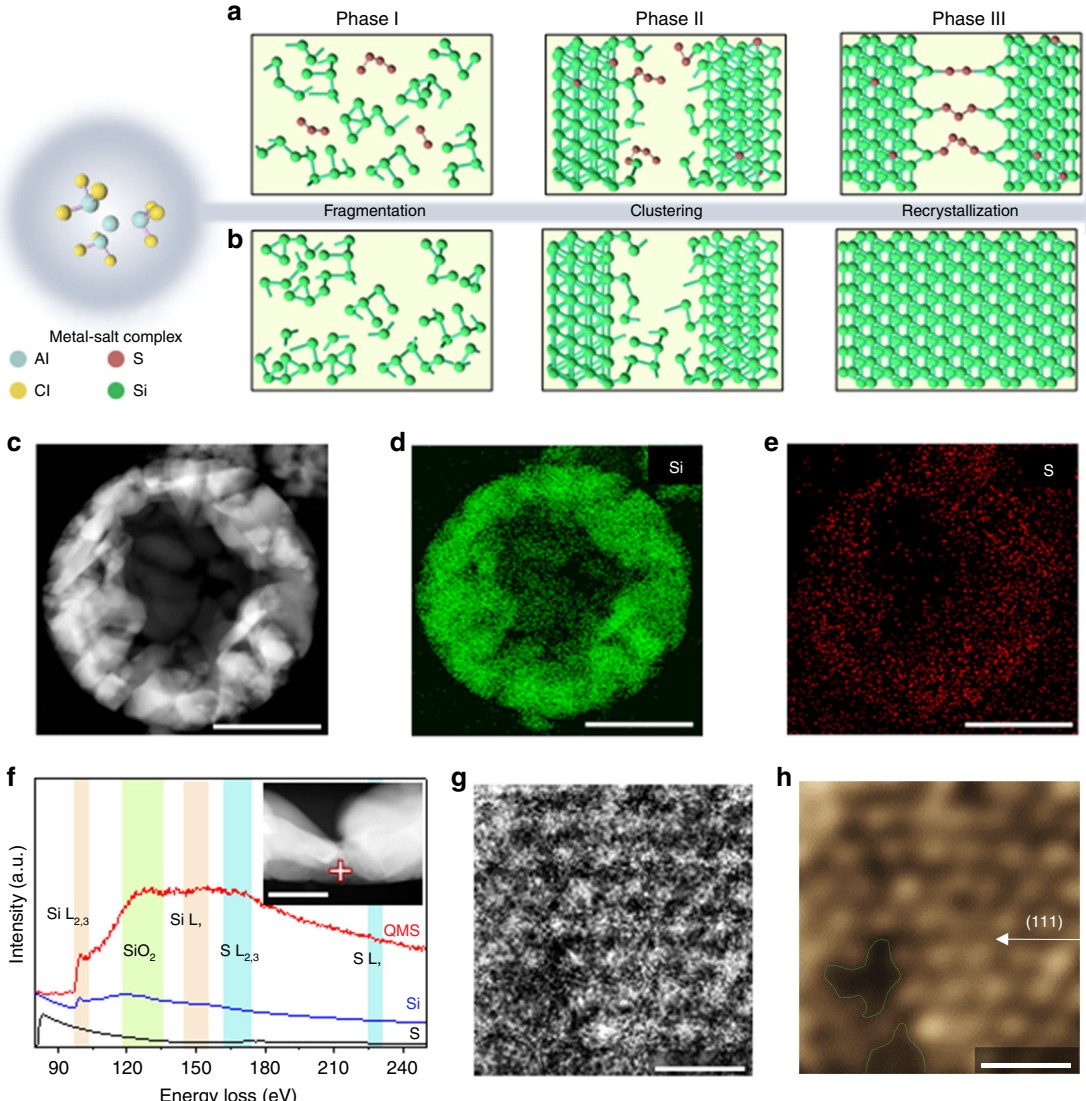

**Fig. 1** Structural evolution and characterization of quasi-metallic silicon. **a**, **b** Schematic of each stage for (**a**) QMS and (**b**) bare Si. **c** Scanning transmission electron microscopy (STEM) image. **d**, **e** Corresponding energy-dispersive X-ray spectroscopy (EDS) maps for (**d**) Si and (**e**) sulfur. **f** EELS spectra for elemental sulfur, bare Si, and QMS (inset: STEM image of the EELS measurement position). **g**, **h** High-magnification STEM images of QMS. Scale bars, 1 μm (**c–e**); 50 nm (**f**); 5 Å (**g**, **h**)

plots for each sample are presented in Fig. 2a. By keeping their microstructures and outer diameters identical, their single-particle conductance can be roughly estimated based on the slopes of the plots. QMS(0.7) exhibits six times higher conductance than undoped Si (Fig. 2b). Hereafter, the number in parentheses in the sample labels refers to doping concentration in atomic percent. Furthermore, a similar circuit was constructed with pelletized samples to measure the conductivity more reliably, and the results were close to the reported conductivity of bulk Si ($0.001 \, \mathrm{Sm^{-1}}$). By contrast, only 0.7 % of sulfur doping dramatically increased the conductivity of Si, reaching up to 50 times higher than that of undoped Si, which confirms the unusual transition to the quasi-metallic state.

Density functional theory (DFT) calculations revealed a correlation between the metallic properties of QMS and sulfur doping by comparing the electronic band structures of Si at different doping concentrations (Fig. 2c, e). After the sulfur fusion, the two remaining valence electrons of Si create two impurity states below the conduction band minimum (CBM), labeled by blue and red lines, and occupy one of the two states.

The localized states are far apart and do not interact, and they just touch the Fermi level at a low doping amount (0.39 at%, inset of Fig. 2c). As the doping concentration increases up to 1.59 at%, these spatial states become closer and overlap (Fig. 2e, inset). This percolation enhances the band dispersion of the states across the Fermi level and forms metallic bands that contribute the metallic property of QMS, which is intensified at concentrations above 0.39 at% (Fig. 2c, e). The impurity levels that lie below the CBM signify the n-type character of QMS, that is also validated in the Hall effect measurements (Fig. 2d and Supplementary Fig. 6). At 1.59 at%, the extended charge density distribution shows overlap between the localized states and the corresponding large band dispersion, indicating the higher electron mobility than that at 0.39 at% (insets of Fig. 2c, e). Thus, the doping concentration of 0.7 at% evidently rendered the Si quasi-metallic.

**Formation of sulfur-supported ion channels**. The sulfur fusion not only enabled the quasi-metallic transition but also produced self-supporting channels within the Si crystal structure, which

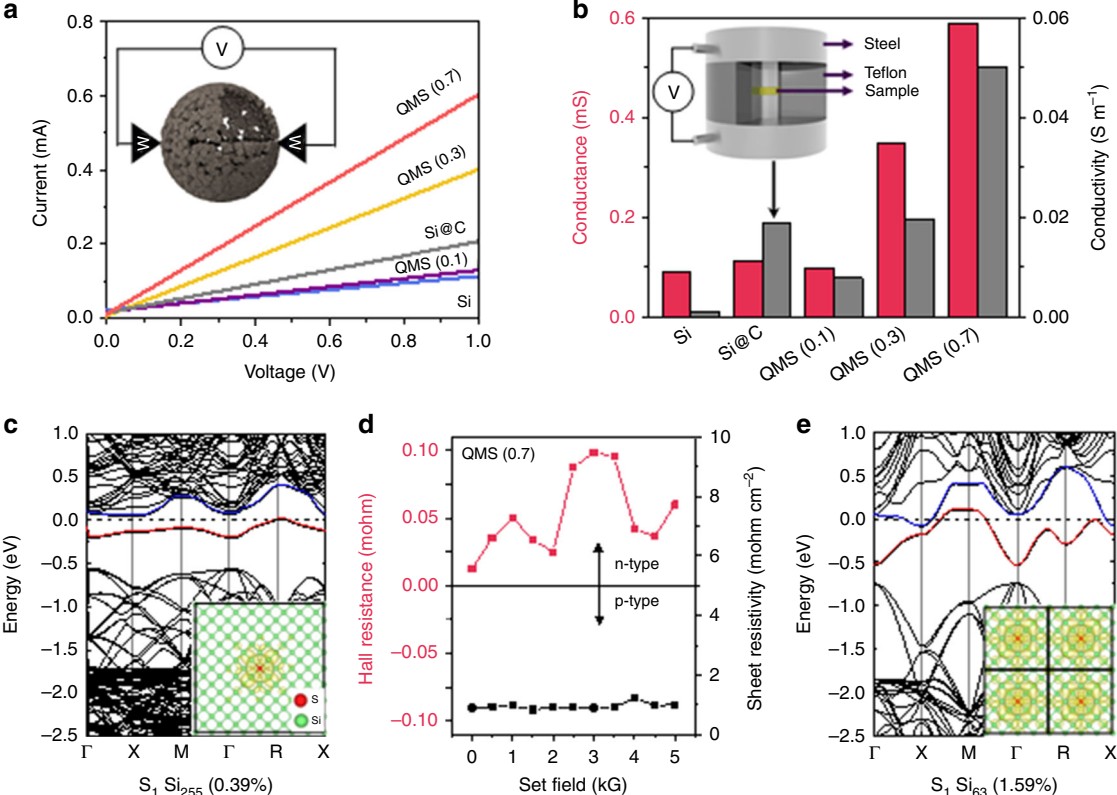

**Fig. 2** Metallicity of quasi-metallic silicon. **a**, **b** Single-particle *I–V* plots (**a**) and bulk (pellet) conductivity results (**b**) of bare Si, carbon-coated Si(Si@C), QMS(0.1), QMS(0.3), and QMS(0.7) samples. **c**, **e** Calculated band structure of QMS samples with different substitution doping concentrations of (**c**) 0.39%, $S_1Si_{255}$, and (**e**) 1.59%, $S_1Si_{63}$. The charge density distribution at these impurity levels (in the insets) is shown in red and blue lines to intuitively illustrate the contribution of Si to the metallic properties. The isosurface of the density is 0.0005 e/Å$^3$. **d** Hall effect measurement results of QMS(0.7)

were distinctively identified by high-resolution transmission electron microscopy (HR-TEM). Along with the defect clouds previously discussed for the substitutional doping, dark strips appear parallel to (111) planes in the polycrystalline Si, which are assumed to be the expanded interplanar spacing from the sulfur chains and are mostly 0.50–0.72 nm wide, in contrast with the 0.31 nm lattice constant for Si (111) planes (Fig. 3a–c). The outmost surface might lose sulfur chains inside the crystal due to the high-temperature post-annealing process, but their traces show the evident formation of robust channels with Si (111) planes bent near the expanded strips and the possibility of internal channel formation beneath the top surface. In contrast to other dopants like boron or phosphorus that lead to a decrease in the lattice spacing with a peak shift to a higher angle[33,34], the sulfur dopants inside crystals offset the lattice contraction by forming the channels (Fig. 3d). Higher doping concentration shows a significant peak shift with a weak shoulder close to the bare Si (111) planes that are entirely opposite to previous observations, thus suggesting a different doping character although same sulfur dopants smaller than Si atoms were used[35].

In the absence of sulfur fusion, a channel with a slab spacing of about 1 nm cannot be maintained because the two separate Si surfaces tend to merge and restore the bulk structure, according to our calculation results (Supplementary Fig. 7a). However, the bridging sulfurs are able to bear the Si planes facing each other and sustain the interplanar spacings; the chain-like sulfur is highly flexible in various configurations and sufficiently robust to support the structure without collapsing at pressures as high as 14 kbar (Fig. 3e). In addition, the most stable channel spacings of 0.46 and 0.81 nm are consistent with the experimental measurements (Fig. 3a–c and Supplementary Fig. 7b, c). The channels can

provide up to 5000 times higher ionic diffusivity than that of bulk Si, as measured by galvanostatic intermittent titration technique at low lithium contents as well as cyclic voltammetry measurement (Fig. 3f and Supplementary Figs. 8 and 9). These results are consistent with the calculated lower diffusion barrier through the channel of 0.11 eV (Supplementary Fig. 7d), in contrast with 0.58 eV through the bulk[36].

**Lithium sulfide-embedded structure.** While the most dopants in single atomic sites hardly achieve a full intercalation of Li compared with its bulk crystal structure and normally considered as dead sites for Li reactions[37,38], the unusual doping nature of QMS will adopt different ways. The fused sulfur in either forms should be discharged (lithiated), at least for the first cycle, with a narrow voltage window that limits subsequent charging (delithiation), thereby indicating the possible formation of lithium sulfide[39]. Through in situ TEM investigation, we evidently found that Li insertion into the QMS particle generates lithium sulfides nanocrystals along with negligible volume expansion from alloy formation of Li and crystalline Si (Fig. 4a–d and Supplementary Movie 1). The appeared sulfur in selected area diffraction pattern might arise from isolation of unreacted sulfur chains and clustering of sulfur atoms in substitutional positions. Importantly, lithium sulfide byproducts remain intact during delithiation in well-trapped forms and will not dissolve in the electrolyte used here for subsequent cycles (Fig. 4e, f, and Supplementary Movie 2) (refs. [40,41]).

Inside the fully amorphized Si structure, rather clustered sulfur particles of <1 nm were observed; otherwise, micropores appeared in these traces during the elimination of the solid-electrolyte

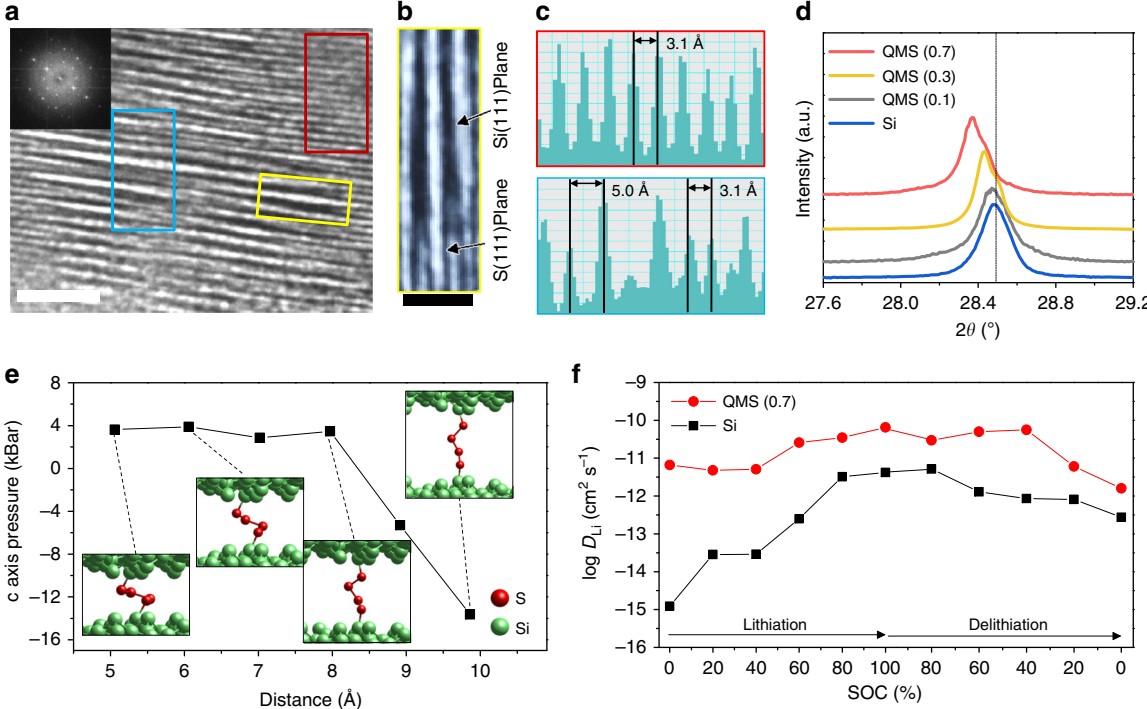

**Fig. 3** Sulfur-fusion-induced channel formation in quasi-metallic silicon. **a** HR-TEM image of QMS (inset: corresponding fast fourier transform image). **b** Enlarged TEM image showing column formation between characteristic Si (111) planes. **c** Intensity profiles of selected areas in (**a**). **d** XRD patterns of Si and a series of QMS between 27.6–29.2°. **e** Sulfur chain structure under applied pressure depending on different channel sizes, as calculated by DFT. **f** Li-ion diffusion coefficient versus the state of charge (SOC) during the first cycle. Scale bars, 2 nm (**a**); 1 nm (**b**)

interphase (SEI) (Fig. 4g, h, and Supplementary Fig. 10). After the first cycle, during which the lithium sulfide particles formed, Li still diffused faster through QMS than undoped Si (Supplementary Fig. 11). We assumed that the lithium-sulfide-related structure sustained the diffusion path, and our DFT calculations found a low diffusion barrier of 0.32 eV (Supplementary Fig. 12a, b). In our calculated model structure, the metallic property of QMS is demonstrated by the change in the CBM occupation (red band) owing to the charge transfer from the lithium sulfide to the amorphous Si. The unoccupied conduction state (red band) of amorphous Si (Supplementary Fig. 12c) is occupied by an electron from lithium sulfide, thereby maintaining its n-type character (Supplementary Fig. 12d). We assert that the metallic property of the QMS containing lithium sulfide nanoparticles mostly arises from the occupied CBM states of amorphous Si via charge transfer from the lithium sulfide. The charge density plots of the CBM state of H-passivated amorphous Si show features similar to those of the metallic state in the lithium sulfide–Si model system, demonstrating fast charge transfer.

**Li storage performance of quasi-metallic silicon**. The sulfur fusion method ensures compelling battery performances in coin-type half- and full cells (the electrode and cell information and electrochemical measurements are described in detail in the Methods section). It is noted that the following discussion on electrochemical behaviors is based on the QMS(0.7) samples unless otherwise noted and those of the QMS(0.3) and QMS(0.1) samples are shown in Supplementary Fig. 13. The quasi-metallic state and the sulfur-buffered Li-ion diffusion channels of QMS increased both the ICE of 92.5% and charge (delithiation) capacity of ~3350 mA h g$^{-1}$, compared with the Si electrode having 87.4% and 3080 mA h g$^{-1}$, respectively in the first cycle at 0.05 C

(1 C=3.5 A g$^{-1}$), thereby suggesting that the particles were almost fully activated (Fig. 5a). As systematically investigated in the literature[42], the conductivity factor much more critically determines the initial reversibility of Li-ion transport in the Si-based anodes with other physicochemical properties being equal than the factor from the surface area of the samples. Further, the highly accessible hollow and porous structures for the electrolyte facilitate quick activation of the entire electrode and the sulfur-buffered Li-ion diffusion channels enhance the diffusion kinetics. The dominant macropores and a minor portion of meso/micropores are also beneficial to achieve the unprecedented high ICE of 92.5% without additional carbon coating layers.

Without the use of typical protective layers, the QMS electrode retained 87% of its capacity over 300 cycles and 72% over 500 cycles at 0.5 C (1 C = 1.9 mA cm$^{-2}$) along with a high average Coulombic efficiency of 99.89% from the 2nd to 500th cycle, thus outperforming previously reported microscale Si anodes (Fig. 5b). In addition to achieving high ICE, quick saturation of Coulombic efficiency during subsequent cycles was challenging in typical bulk Si anodes due to sluggish diffusion of Li-ion through the huge structure. However, highly accessible structure toward the Li-ion diffusion as well as metallicity of QMS contributes to having 99.5% of Coulombic efficiency at the third cycle even at a relatively slow rate of 0.5 C, which renders the QMS electrode suited for the realization of full cell. At this high-capacity loading of ~3.8 mA h cm$^{-2}$, it retained its metallic nature and stabilized interfaces, which facilitated fast Li-ion kinetics for unprecedented bulk rate performance when the current densities were increased up to 5 C (19 mA cm$^{-2}$) without any trace of lithium metal plating. This result was corroborated by in situ electrochemical impedance spectroscopy which showed that both undoped Si and QMS delivered almost 100% of available capacity at 1 C, while the average charge transfer resistance of undoped Si was higher than

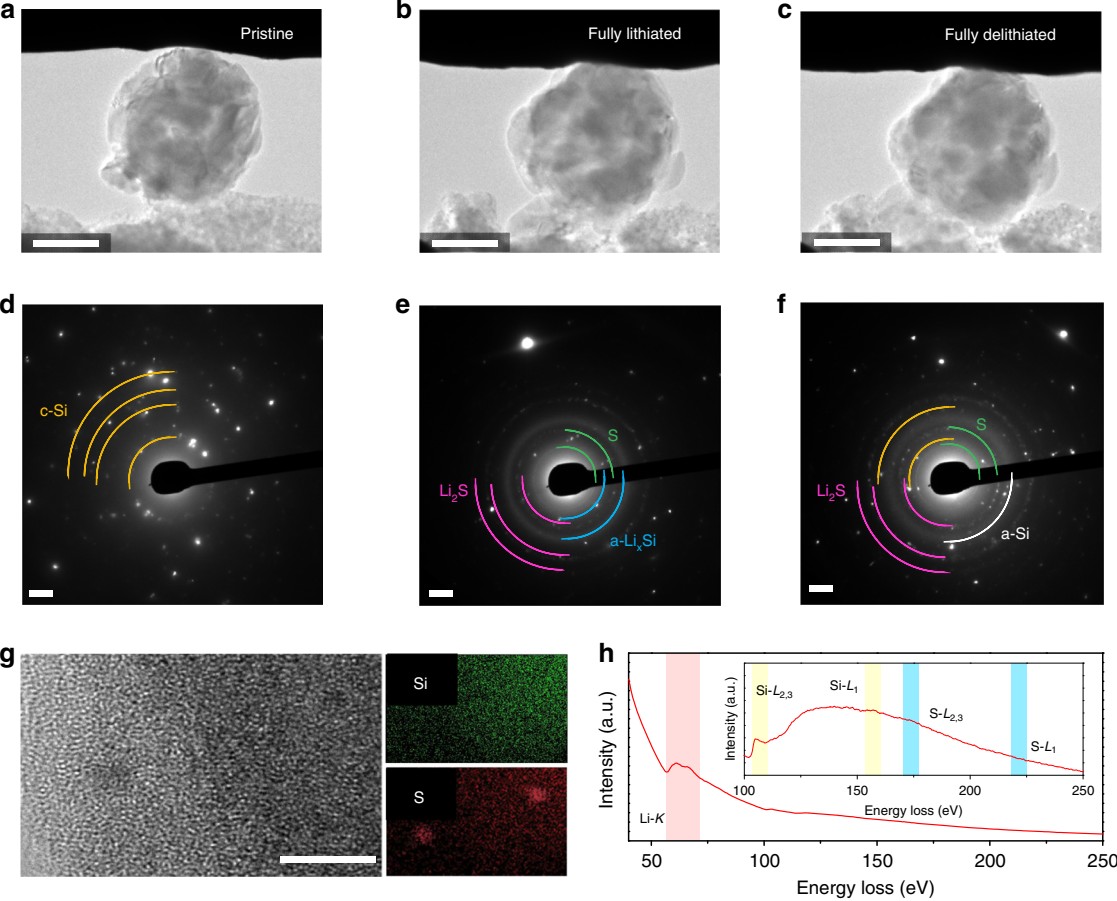

**Fig. 4** Lithium sulfide formation inside quasi-metallic silicon. **a–f** TEM images, and corresponding selected area diffraction patterns for (**a**, **d**) pristine, (**b**, **e**) fully lithiated, and (**c**, **f**) fully delithiated states of QMS particle. **g**, **h** HR-TEM image and corresponding elemental maps for Si and S (**g**) and EELS spectrum (**h**) of QMS after cycles and SEI elimination. Scale bars, 500 nm (**a**, **c**, **e**); 2 1/nm (**b**, **d**, **f**); 5 nm (**g**)

that of QMS due to the retained metallicity (Fig. 5c and Supplementary Figs. 10 and 14−16). The QMS electrode could fill more than 70% of the initial available capacity with the low charge transfer resistance at 3 C, but <30% of the capacity was obtained with the increased resistance in case of undoped Si, suggesting that the QMS can be fast-chargeable even with the micrometer size which is essential for the further applications[43,44].

A sufficient but not immoderate interior pore volume inside QMS can sustain repeated large volume expansion by remaining as low as 50% and relieving generated internal stress generated by Li insertion, thereby improving the fracture resistance on the particle level[21] (Supplementary Figs. 17 and 18 and Supplementary Videos 1 and 2). The robust microparticle structure significantly suppressed electrode swelling to <30% after 100 cycles, which corresponds closely with volumetric margins in industrial cells (Fig. 5d and Supplementary Fig. 19) (ref. [11]). By fulfilling the rigorous requirements for practical full cells such as structural stability and reaching a high Coulombic efficiency during early cycles, the QMS anode in a finite source of Li ions using traditional lithium cobalt oxide (LCO) exhibited stable cycling (200 cycles, 80% capacity retention) at a high current density of 3.3 mA cm$^{-2}$ and an areal capacity loading of ~3.3 mA h cm$^{-2}$ (Fig. 5e and Supplementary Figs. 20 and 21). The decreased polarization from the metallic nature of QMS and the partially retained ionic channels during cycling lead to a high volumetric/gravimetric energy density of full cells compared with other promising designs for Si-based anodes (Supplementary

Table 1 and Supplementary Note 2), which can potentially be further increased by developing cathodes that is stable at a fast-charging rate with high-energy density.

## Discussion

Doping <1% sulfur into the Si structure modifies the physical properties of bulk Si by imparting it with an electronically conductive state and creating ionic channels for fast Li-ion diffusion. Such doping is enabled by a safe, scalable, and feasible approach, in contrast with conventional high-risk and toxic methods. In addition to sulfur dopants, other chalcogens including selenium or tellurium are expected to induce insulator-to-metal transition in the same way but with different critical doping concentrations as reported previously[45,46]. Whether these larger atoms form the chain-like structure that creates diffusion channels for metal ions raises an open question; otherwise, the air-stable binary phase will appear rather than simultaneously featuring metallicity and channel formation[47]. These unusual doping characteristics can address the major issues of Si anodes, which mostly arise from their large volume expansion during electrochemical cycling at high mass loading and in the absence of conductive buffer layers. The metallic nature at the interfaces of the amorphized Si and lithium sulfide maintains the electronic and ionic conductivities of the microparticles, and the porous structure prevents particle disintegration and severe electrode swelling, thereby extending the cycle life of batteries while maintaining a high-energy density. The low-temperature sulfur fusion proposed here may advance Si-based

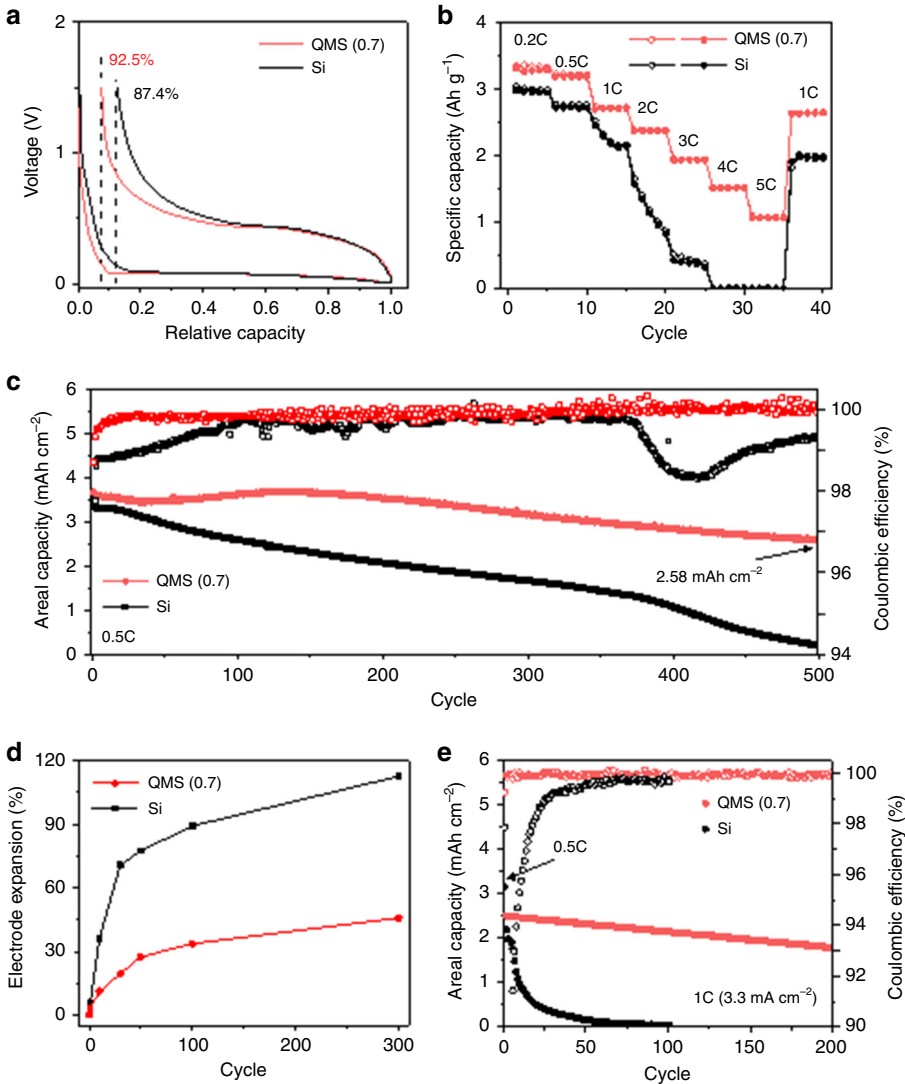

**Fig. 5** Li-ion storage properties of quasi-metallic silicon and Si electrodes. **a** First charge–discharge profiles of QMS(0.7) and Si electrodes at a C-rate of 0.05 C. **b** Charge areal capacity retention of both electrodes at 0.5 C with the corresponding Coulombic efficiency. **c** Specific capacity plots for both electrodes at different C-rates from 0.2 C to 5 C. **d** Electrode expansion ratio during 300 cycles for both electrodes. **e** Full-cell cycle retention at 1 C (3.3 mA cm$^{-2}$) along with the coulombic efficiency

high-energy batteries in practical systems and open a new paradigm in the design of materials for various electronic applications.

## Methods

**Materials and characterization**. Micrometer-sized silica (1 μm, 99.9%), magnesium sulfate (anhydrous, 99.5%) and aluminum chloride (anhydrous, 99%) were purchased from Alfa Aesar. Aluminum metal (1–5 μm, 99%), hydrochloric acid (35–37%), and hydrofluoric acid (49%) were purchased from Angang, SAMCHUN, and J.T. Baker, respectively. All the chemicals were used without further purification. The structural analysis was carried out by field-emission SEM (S-4800, Hitachi) with an acceleration voltage of 10 kV. HRTEM observations were conducted on Titan 80–300 environmental TEM (FEI) and field-emission TEM (JEM-2100F, JEOL) with EDS detector operated at 300 and 200 kV, respectively. Crystal structure of samples were characterized by X-ray diffractometer (XRD, Bruker D8-advance) at 3 kW using Cu Kα radiation in the θ range from 20° to 90° and confocal Raman (alpha 300 R, WITec) with 532 nm of wavelength laser. The surface area and pore size distribution were characterized by auto physisorption analyzer for BET and BJH analysis (ASAP 2020, Micromeritics Instruments). The XPS (K-alpha, ThermoFisher) analysis was used for the surface oxidation state of the samples. For bulk conductivity measurement, the same amounts of the samples are poured into the steel cylinder with an area of 1 cm$^2$ and height of 1 mm with an upper and lower connection of external circuits. The hall effect of the samples was measured with the Hall measurement system (7770A Lakeshore, Bipolar electromagnet) and electrodes casted onto the polyethylene terephthalate film substrate without conductive carbons.

**Quasi-metallic silicon synthesis**. In a typical synthesis of QMS samples, SiO$_2$, MgSO$_4$, Al metal, and AlCl$_3$ were finely ground with a mass ratio of 1:0.5–3:2:10 using an agate mortar and transferred to stainless steel reactor consisting of one union and two plugs inside an Argon-filled glove box. After fastening the reactor securely, it was transferred to the tube furnace and heated at 250 ℃ for 10 h under argon atmosphere. After complete cooling, the product looked like hard and rigid rock, but it is easily swelled out with water treatment to dissolve excessive AlCl$_3$ salts and remove the undesirable silicon sulfides. Then, the intermediate consisting of QMS, residual Al metals, AlOCl byproducts, and elemental sulfur were purified with 1.0 M HCl and 5% HF, respectively. Note that the crude products smelled similar to that of a rotten egg that implies a formation of sulfur derivatives from MgSO$_4$. Through additional heat treatment at 400 ℃ for 30 min under an argon atmosphere, any chance of existence of elemental sulfur was eliminated. The Si samples were prepared without the use of MgSO$_4$ and additional heat treatment step.

**Electrochemical measurement**. A slurry coating method was used to prepare the working electrodes by mixing the anode materials, super P carbon black, polyacrylic acid (weight-average molecular weight = 10 kg mol$^{-1}$, Sigma-Aldrich) and carboxymethyl cellulose sodium salt (Sigma-Aldrich) with a mass ratio of 80:10:5:5 and casting the slurry on copper foil without further calendaring process. The mass loading of anode materials excluding super P and binders maintained 1–1.1 mg cm$^{-2}$. In addition, LiCoO$_2$ (LCO, LG Chem) cathodes were prepared by mixing with super P carbon black and polyvinylidene fluoride (PVdF) binder in a mass ratio of 95:2.5:2.5 and casting the slurries on aluminum foil. The mass loading of cathode

materials excluding super P and binders reached ~23 mg cm$^{-2}$ with an areal capacity loading of ~3.3 mA h cm$^{-2}$, respectively. After casting, the electrodes were completely dried at 150 °C for at least 2 h under vacuum. The prepared electrodes were cut into discs and assembled into the CR2032 cells (Welcos) in an Argon-filled glove box using a Celgard 2400 separator, Li metal counter electrode and electrolytes dissolving 1.3 M LiPF$_6$ in ethylene carbonate (EC) and diethyl carbonate (DEC) (3:7 v/v) with 10 wt% of fluoroethylene carbonate (FEC) additive to increase the cycle life of the battery. One hundred and twenty microliters of electrolyte was used for a single coin cell assembly. The galvanostatic battery tests on the anodes were carried out with the cut-off voltage of 0.005–1.5 V vs. Li$^+$/Li for formation cycle at 0.05C and 0.01–1.2 V vs. Li$^+$/Li for subsequent cycles at 0.2–5C, respectively, on a battery cycler (WBCS 3000K8, Wonatech). For cathode half cells, cut-off voltages are 3.0–4.3 V for LCO. All the specific capacities were calculated based on the mass of Si only. The presented capacities, initial Coulombic efficiency, capacity retention, and rate capability results were collected from at least five cells. The n/p ratio (the capacity ratio of the anodes to cathodes) was ~1.1. The cut-off voltage for full cells were 3–4.2 V for QMS (or Si)-LCO full cell. The CV measurements were obtained at 0.1–1.0 mV s$^{-1}$ from 0 to 1.2 V (VMP3, Biologic). The EIS measurements were carried out between 100 kHz and 0.1 Hz with an amplitude of 10 mV. The impedance spectra of QMS/Si half cells were measured by galvanostatic electrochemical impedance spectroscopy (in situ EIS) analysis. The measurement was carried out after the cells were stabilized by formation cycle at 0.05C. The input signals were combining the sinusoidal alternating current waves of amplitude as low as 10 μA at $10^{-3}$ to $10^6$ Hz and fixed direct current of 1C or 3C. One potentiostat channel (VSP300, Biologic) was used for measuring impedance spectra and the other for recording voltage profiles of the cells.

**In situ transmission electron microscopy analysis**. The in situ TEM measurements were performed in Titan 80-300 environmental TEM (FEI) at the acceleration voltage of 300 kV using a dual-probe electrical biasing holder (Nanofactory Instruments). The QMS (or Si) particles were drop-cast onto a gold wire as a working electrode, while a piece of lithium was attached to a tungsten rod served as the counter electrode. During the transfer of the holder into the TEM, the Li metal was shortly exposed to air for about 3 s to create a thin Li$_2$O layer acting as a solid electrolyte. A constant potential of ±3 V was applied to the QMS (or Si) electrodes against the Li metal during lithiation and delithiation, whereby Li-ion was inserted and extracted through the solid electrolyte. It does not make sense for us to estimate the ion conductivity of particle itself during in situ TEM analysis, because it was measured using the lithium oxide solid electrolyte and each measurement had a different voltage bias depending on its size and contact features. Nevertheless, the QMS particles can be lithiated faster than that of Si even with a lower bias of −3 V as compared with our previous report[7].

**Calculational methods**. Ab initio calculations were performed using the Vienna ab initio simulation package (VASP) code[48] in the framework of the spin-polarized density functional theory with the projector augmented wave (PAW) method[49]. The exchange-correlation was considered using the generalized gradient approximation of Perdew, Burke, and Ernzerhof (PBE)[50]. The cut-off energy for the plane wave basis set was 350 eV. A k-point mesh in the Monkhorst-Pack scheme[51] was set to 1 × 1 × 2 and 2 × 2 × 2 for S doped Si (S$_1$Si$_{255}$ and S$_1$Si$_{63}$, respectively) and 1 × 1 × 1 for the channel structure. The ionic positions of all atoms were fully-relaxed until a force convergence of 0.01 eV Å$^{-1}$ was reached. The pressure applied to the channel depending on the different slab spacing was calculated to be set to allow only ion relaxation without volume change.

Density functional molecular dynamics (DFTMD) simulation was conducted on a canonical ensemble to generate amorphous Si for the interface structure with lithium sulfide particles. The k-point set was set to only the gamma points for 2 × 2 × 2 supercell of Si and the time step was set to 0.5 fs. The temperature was chosen as 1800 K, and the DFTMD simulations were performed for 2.5 ps. To determine the kinetic behaviors of Li-ion in the channel and the interface, we used the climbing image nudged elastic band (cNEB) method[52] to calculate the diffusion barriers of Li-ion on expected diffusion pathways.

## Data availability
All relevant data supporting the findings of this study are available within the paper and its Supplementary Information. Additional data are available from the corresponding author upon request.

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

## Acknowledgements

This work was supported by the Center for Advanced Soft-Electronics funded by the Ministry of Science, ICT and Future Planning as Global Frontier Project (CASE-2015M3A6A5072945), and the support from NRF (2017R1A4A1015323, 2017R1D1A1B03028004). C.M.W. thank the support of the Assistant Secretary for Energy Efficiency and Renewable Energy, Office of Vehicle Technologies of the U. S. Department of Energy under Contract No. DE-AC02-05CH11231, Subcontract No. 18769, and No. 6951379 under the Advanced Battery Materials Research (BMR) program. The microscopic analysis in this work was conducted in the William R. Wiley Environmental Molecular Sciences Laboratory (EMSL), a national scientific user facility sponsored by DOE's Office of Biological and Environmental Research and located at PNNL. PNNL is operated by Battelle for the Department of Energy under Contract DE-AC05-76RLO1830.

## Author contributions

J.R. and J.H.S. contributed equally to this work. J.R. and S.P. conceived the concept. J.R. designed and carried out the experiments, physical characterization and electrochemical test. G.S. and C.W. performed the in situ TEM characterization and data analysis. D.H. contributed to physical characterization and data interpretation. J.H.S., K.C., H.L., and J.H.L designed the modeling and simulations. J.R. and J.H.S. wrote the manuscript. All authors discussed the results and commented on the manuscript.

## Additional information

**Competing interests:** The authors declare no competing interests.

