## [Peer Review File · Nature Communications]

Reviewers' comments:

Reviewer #1 (Remarks to the Author):

This is a high quality paper well written and analyzed for materials characteristics and their electrochemical properties as promising high capacity anode for lithium-ion batteries. With the synthetic protocols reported here, a small amount of sulfur is doped during the bottom up reduction process of silica to form a quasi-metallic porous silicon structure which is turned out to be well accommodating volume expansion during cycles leading to high cycling stability. All these are supported by enough data in the manuscript as well as in the supporting data. The new results reported here will attract broad scientific and industrial interests in the field of energy storage via lithium-ion batteries. Hence, I would like to support the acceptance of current version of manuscript. Before it is accepted, however, the authors are recommended to consider the followings.

1. The authors are recommended to explain the sulfur doping mechanism in detail for potential readers.
2. The QMS itself showed stable cycling without any additional carbon coating process. Is it claimed that the QMS is used as it is as anode for LIB anode?
3. How is the thermal stability of QMS at high temperature in air or in inert gases?

Reviewer #2 (Remarks to the Author):

This manuscript demonstrates that doping of silicon with less than 1% sulfur modifies the physical properties of the bulk Si by imparting it with an electronically conductive state and creating ionic channels for Li-ion diffusion. Furthermore, the doped silicon shows somehow improved electrochemical performance in lithium half and full cells. Some of the results are interesting, however, the work provides limited insight into the silicon anode issue and the battery field. Specific comments:

1. As some chalcogens including selenium or tellurium have been demonstrated to hold the ability of inducing insulator-to-metal transition of silicon, the sulfur doping of silicon in this work and its ability to induce insulator-to-metal transition is not originally and conceptually novel.
2. With excessive structural discussion that could be moved to SI, the QMS remains unclear.
3. Although the increased electronic conductivity of the QMS is observed as shown in Figure 2, there is not any supportive evidence to correlate this increased electronic conductivity and the electrochemical performance, since the conversion of sulfur toward lithium sulfide upon cycling. That is, the section demonstrating electronic conductivity increase has little relationship with the main conclusion of the manuscript.
4. The authors overstated the role of sulfur or lithium sulfide in electronic and ionic conductivity; the comparison and discussion were excessively focused on the role of sulfur or lithium sulfide. In the referee's opinion, other factors, for example, the pore structure, may mainly contribute to the observed difference. Yet, given the pore structure modulation, the novelty of the work is limited again while pore engineering has been widely demonstrated.
5. The QMS shows relatively improved performance compared to its counterpart provided in this work. Yet, the QMS performance either in half cells and full cells is rather mediocre, and the significant capacity decay was still observed upon cycling.
6. The authors show improved ICE, yet the related explanation is missing. In fact, it is very strange for the referee to see this improvement without any interface modification.

Overall, the referee has concerns about its novelty, importance and the impact on this field. Thus, it is hard for me to recommend publication of this manuscript in Nature Communications.

Manuscript ID: NCOMMS-19-00539-T

Responses to all reviewer comments for Manuscript ID: NCOMMS-19-00539-T

(**Title:** Infinitesimal sulfur fusion yields quasi-metallic bulk silicon for stable and fast energy storage)

Reviewers' comments as follows;

Reviewer# 1

Comments:

*This is a **high quality paper** well written and analyzed for materials characteristics and their electrochemical properties as promising high capacity anode for lithium-ion batteries. With the synthetic protocols reported here, a small amount of sulfur is doped during the bottom up reduction process of silica to form a quasi-metallic porous silicon structure which is turned out to be well accommodating volume expansion during cycles leading to high cycling stability. **All these are supported by enough data in the manuscript as well as in the supporting data.** The new results reported here will **attract broad scientific and industrial interests** in the field of energy storage via lithium-ion batteries. Hence, I would like to support the acceptance of current version of manuscript. Before it is accepted, however, the authors are recommended to consider the followings.*

Response: We thank the reviewer for the positive comment and comprehensive understanding of our work with constructive comments, which are addressed in this round of revision per attached below. The revised or newly added sentences in the revised manuscript are highlighted as red text.

Comment 1-1) *The authors are recommended to explain the sulfur doping mechanism in detail for potential readers.*

Response 1-1: We appreciate the reviewer's positive comments on our manuscript which will strengthen our manuscript. We recognize that proposed doping mechanism in this original article might be less specific for the non-specialists (potential readers) in other fields as the reviewer pointed out. In fact, the proposed mechanism of sulfur dopants in the bulk Si materials is based on

our previous report (*Commun. Chem.* **1**, 42 (2018)) that fundamentally demonstrated how the metal and metal salts form a complex structure, adsorption conformations and reduction activity of these species. Also, our ‘solid experimental evidences’ support such speculations. At well above the melting point (T_m) of metal salts (AlCl_3 , $T_m \sim 192^\circ\text{C}$), which is significantly lower than the required thermal energy of typical thermochemical reduction process of silica (e.g., Mg for 650°C and Al for 660°C), these salts melt to form a molten salt medium inside a closed reactor and then solvate the Al metal at low temperature to make a Al- AlCl_3 complex. Afterward, activated AlCl_3^* from the ligand of the complex in two different thermodynamic pathways is spontaneously adsorbed onto the oxygen atoms of silica, thus producing a crucial byproduct of aluminum oxychloride (AlOCl) and creating the Si-Si/S-S/Si-S bonds. The successive reduction reactions provide a clustering environment of Si seeds and they are eventually recrystallized into spherical shell structure via localized Ostwald ripening process, while the byproducts and excessive molten salts contribute to macro-/meso-pores of hollow and porous Si. Importantly, the final morphology derived from this low-temperature reduction reaction does not depend on the morphologies/purity/chemical compositions of initial precursors (SiO_2 , clays, etc.), in which the same manner will apply in our S-doped Si systems and accordingly our approach features mass-scalability and versatility.

Evidently, we observed S-related products after the reduction reactions of sulfate salt precursors in two different formulations (one for sulfate reduction and the other for silica/sulfate reduction in presence of Al/ AlCl_3) as shown in **Supplementary Figure 1**; XRD patterns confirm the formation of elemental sulfur reduced from the sulfate by Al- AlCl_3 . This result implies that high oxygen affinity of the complex (toward electron-rich elements such as oxygen, sulfur, etc.) will lead to a strong adhesion to O-edged surface of salt crystal (MgSO_4), whereby isolates (or creates) a sulfur seed along with formation of Mg-Al-Cl complex (MgAl_2Cl_8), and relatively small amounts of AlOCl as reported previously (*Energy Environ. Sci.* **8**, 3187 (2015)). Thus, during the reduction process in the presence of both sulfate and silica, two different seeds will be formed and give a uniform mixture in the molten salt media, then finally sulfur seeds are uniformly doped into the Si host structure in micrometer scale unlike the surface-rich doping from the ion-implantation and destruction of the original structure.

In detail, we need to pay attention to the formation of MgAl_2Cl_8 which was known as byproducts of the reaction in presence of metallic magnesium and AlCl_3 (e.g., $2\text{Mg} + \text{SiO}_2 + 6\text{AlCl}_3 \rightarrow 2\text{MgAl}_2\text{Cl}_8 + 2\text{AlOCl} + \text{Si}$). This implies that, as we assumed and even believed in this work, Al- AlCl_3 complex could selectively reduce MgSO_4 and then produces metallic Mg, amorphous sulfur (sulfur seeds) and AlOCl byproducts. Accordingly, the produced Mg will participate in subsequent reduction reaction as following the above equation and generate an unexpected product of MgAl_2Cl_8 . In other words, we revealed, for the first time, that high reactivity of the complex can reduce the ceramics (metal oxides) and even oxygen-containing salt crystals, thus demonstrating that our versatile approach can be extended to introduce the dopants

in to the host structure from the seed level and yield uniformly doped materials according to specific purpose.

In order to clarify the reduction mechanism in this study, we added the following description and supplementary information to the revised manuscript as follows (Page 4, line 69-89);

“In principle, aluminum chloride (AlCl_3) used here as metal salts (T_m of 192°C) as well as molten salt medium can solvate bulk aluminum (Al) to form the highly reactive Al-AlCl_3 complex structure that spontaneously reduces different types of silica (e.g., cost-effective clay minerals and commercial bulk SiO_2) in two thermodynamically stable pathways. Activated AlCl_3^ from the ligand of the complex adsorbs on the oxygen atom of these compounds and then generates unusual byproducts of aluminum oxychloride (AlOCl) along with the formation of Si seed as demonstrated previously⁷. The complex species also react with an additional metal salt (MgSO_4), selectively dissociating the oxygen atoms from the salt crystal structure and thereby yield isolated magnesium (Mg) and sulfur. The presence of metallic Mg, which acts as the metal center of the complex, leads to the evolution of secondary byproducts of MgAl_2Cl_8 as clearly evidenced by X-ray diffraction (XRD) analysis of crude products after the reduction reactions (**Supplementary Fig. 1**). Further, protuberant XRD peaks observed at near 20° and 40° correspond to the amorphous sulfur clusters while still buried with sharp peaks for either metallic Al or Si, which are indicative of the simultaneous formation of Si and sulfur seeds.*

Strong reducing power of the complex completely disintegrate initial precursors into the active Si and sulfur at atomic level as suggested above but still stabilized by and embedded in the fluidic molten salt medium. Afterwards, abundant byproducts set the clustering environment for each atom to grow into the seeds that have a relatively free motion in the medium and eventually they are assembled into the spherical structure to reduce the surface energy of particles as the reactor cools down to ambient temperature.”

Supplementary Figure 1 | Seed formation analysis. a, b XRD patterns of crude products and samples after H₂O wash in (a) MgSO₄-Al-AlCl₃ and (b) SiO₂-MgSO₄-Al-AlCl₃ systems with other reaction parameters kept identical described in the experimental section.

Comment 1-2) *The QMS itself showed stable cycling without any additional carbon coating process. Is it claimed that the QMS is used as it is as anode for LIB anode?*

Response 1-2: We appreciate the reviewer's comment which will strengthen our manuscript. As the reviewer grasp the bottom line of this work well, the prepared QMS here was evaluated as an anode material for LIB without any additional carbon or buffer layers which have been extensively studied in the literature. In general, Si materials could intake large amounts of lithium ions up to 3.75 Li⁺ per unit structure and this leads to large volume expansion/contraction during repeated charge-discharge cycles, which in turn raised critical issues such as particle pulverization, electrode delamination, and unstable SEI layers. Several strategies proposed to mitigate structural instability, such as nanostructuring, the unusual structure of yolk-shell/core-shell, the introduction of buffer layers, and multifunctional binders as described in the introduction of the original manuscript. However, most of them are not suited for the practical purpose due to high cost, complicated preparation set-up, and poor material density. Alternatively, there have been few attempts to utilize Si microparticles under the consideration of practical feasibility; coalescing the particles inside conformal graphene cage (*Nat. Energy* **1**, 15029 (2016)), self-healing binders (*Nat. Chem.* **5**, 1042 (2013)), highly elastic binders (*Science* **357**, 279 (2017)) based on molecular pulley's action, and etc. Despite important breakthroughs from such designs, no one has achieved stable Si structure in micrometer scale without the buffering and supporting materials.

Remarkable electrochemical performances obtained from QMS anode can be attributed to three essential distinctions as follows:

- i) Resulting QMS itself has a fracture-resistant structure upon the huge stress generation because of abundant meso-/macropores and internal void spaces in spite of its bulk size (*ca.* 3 μm) as primarily demonstrated in our previous work (*Commun. Chem.* **1**, 42 (2018)). Interestingly, this unique structure even with uniform sulfur doping was enabled by one-pot low-temperature reduction reactions. As shown in the porous and hollow nano-Si (inner and outer diameter of 200 nm and 350 nm, respectively) anode (*Nat. Commun.* **6**, 8844 (2015)), we observed similar inward breathing of structure during lithiation in this bulk Si materials (QMS) and reversible accommodation of volume changes. This implies that ensured structural stability of hollow/porous Si microparticles leads to interfacial stability as well.
- ii) Sulfur atoms doped into the substitutional positions in QMS significantly increased electric conductivity by 50 times compared with that of undoped Si, which is related to the insulator-to-

metal transition of typical chalcogen-doped Si. We carefully validated its conductivity (or conductance) in three different ways such as bulk conductivity based on the pelletized samples, single particle conductance from in situ measurement, and Hall effect measurement. Typical dopants of boron or phosphorous to Si materials will increase electric conductivity as high as what we can attain from QMS samples, but they require higher amounts of doping concentration (>3 at%) while available capacity of doped Si with B/P elements will be significantly lowered (*ACS Appl. Mater. Interfaces* **8**, 7125 (2016)). However, here with only 0.7 at% of sulfur fusion into the bulk Si materials, we could achieve good electric conductivity even higher than the carbon-coated Si. Also, minimized amounts of dopants increase available capacity during cycles. Importantly, the doping effect on Si anode was similar to other reports such as improved ICE (> 92%) and fast activation of particles (CE of 99.5% at the 3rd cycle).

iii) In terms of ion diffusion deep into the structure, we observed sulfur chain-supported ion channels inside the Si structure. Such buffering chains are robust and flexible to sustain the original structure of Si without alteration on crystallinity and physicochemical properties of Si. The sulfur chains will readily react with lithium ions to generate lithium sulfide nanoparticles (<2 nm) as demonstrated by both in-situ and ex-situ microscopy observations. From our theoretical calculations, we found that the lithium-sulfide-related structure sustained the fast diffusion path and QMS samples retained metallic character at the interface of amorphized Si and lithium sulfides, which help to enhance the lithium-ion diffusions as well as overall electrochemical properties during cycles.

Based on the above three major advantages of our approach, we render QMS anodes more stable and fast-chargeable without the need for carbon coating layers. (Please note that additional carbon coating on QMS will certainly enhance the electrochemical performances far beyond state-of-the-art battery anodes but this is not consistent with our report and will rather interrupt solid understanding of QMS samples first reported here)

Comment 1-3) *How is the thermal stability of QMS at high temperature in air or in inert gases?*

Response 1-3: We appreciate the reviewer's comment which will strengthen our manuscript. As we described in the Method section, the mild thermal treatment at 400 °C for 30 min was conducted to eliminate any residual elemental sulfur inside the structure by chance. In fact, the leaching process with HCl/HF just before the annealing successfully removed amorphous sulfur residues which are not embedded in the structure and has shown almost identical electrochemical performances without additional heat treatment.

However, as the review pointed out, sulfur compounds have been usually considered thermally unstable in both air and inert environments since the T_m of sulfur is about 115 °C.

Therefore, we additionally annealed QMS(0.7) samples at 700 °C for 1 h (batch: 0.1 g) to check its thermal stability as well as corresponding electrochemical performances. Up to this temperature, Si materials are assumed to be not severely oxidized as reported in the literature (*Electrochim. Acta* **174**, 688 (2015)). Simply, we check the crystallinity from XRD measurement (**Figure R1**), which shows almost same XRD patterns matched to the typical cubic crystal structure of Si (ICSD no. 01-089-5012), while first-order peaks of Si(111) plane for the samples is different each other. When annealed in the inert gas (Ar), protuberant shoulder peak which appears in the pristine sample (QMS(0.7)) disappears but it still remains left-shifted, suggesting that strong bonding of doped sulfur and Si partially sustain even at the harsh conditions. Also, a higher full-width half-maximum value than that of undoped Si samples is ascribed to the slightly enhanced crystallinity from the high-temperature annealing process (*J. Electrochem. Soc.* **136**, 1169 (1989)). However, the peak for annealed samples in the air shifts to the right and becomes broader than others (*Nanoscale Res. Lett.* **13**, 134 (2018)). The sulfur-doped sites might be considered as defect sites which is less stable and easily attacked by the reactive gas like oxygen, thus possibly increasing the oxygen contents by facile diffusion of oxygen molecules through the defect sites. In other words, QMS samples are relatively stable even at the high-temperature treatments, while oxygen gas will aggressively react with a sulfur dopant to diminish doping effects. Compared with commercial Si particles which have been already passivated well with native oxide layers (< 2 nm), QMS samples after annealing in the air will have a much thicker silicon oxide layers due to uniformly distributed defect sites from sulfur doping.

Galvanostatic measurements of these treated samples were carefully carried out in the same cell architecture (2032 coin-type cell with same separator and electrolyte used in the manuscript) and electrode composition (active materials/binder/super-P=80:10:10). As inferred from the XRD results, QMS(0.7)-Ar electrode has a relatively lower charge (lithiation) capacity of 2,929 mA h g⁻¹ and initial Coulombic efficiency (ICE) of 89.1% compared with those from QMS(0.7) electrode. The oxidized sample delivers a higher discharge capacity of 3,332 mA h g⁻¹ but its reversibility (ICE~81.8%) is quite poor, which originates from the undesirable/irreversible lithium consumptions based on the reaction between excessive silica on the surface and lithium to form a lithium silicate (**Figure R1c**). Such irreversible loss from the formation of insulating byproducts closely correlates with the above XRD results. The decreased electric conductivity of the thermally treated samples and possible collapse of ionic channels inside the structure might cause the above differences. In the subsequent cycling, QMS(0.7)-Air electrode retains less than 50% of its discharge capacity after 50 cycles at 1 C, while the QMS(0.7)-Ar electrode still delivers 70% of the capacity which is similar to the results from QMS(0.1) electrode but with lower reversible capacity (**Figure R1d**).

We considered adding this discussion on thermal stability of the prepared samples to the revised manuscript, but we have already conducted mild thermal treatment at the end of samples preparation and also found that the series of QMS samples did not lose its unique property even

when stored in ambient environment as well as thermally treated at 400 °C, which did not cause any deviations on physical and electrochemical properties. Thus, we would like to suggest leaving this discussion and related results in this revision only.

Figure R1. Analysis on thermal stability of QMS samples. (a,b) XRD patterns of undoped Si, QMS(0.7) and thermally treated in Ar and Air environments. (c) Voltage profiles of those thermally treated samples at the first cycle and (d) cycle retentions. Areal loading level was similar to that used in this work.

Reviewer# 2

Comments:

This manuscript demonstrates that doping of silicon with less than 1% sulfur modifies the physical properties of the bulk Si by imparting it with an electronically conductive state and creating ionic channels for Li-ion diffusion. Furthermore, the doped silicon shows somehow improved electrochemical performance in lithium half and full cells. Some of the results are interesting, however, the work provides limited insight into the silicon anode issue and the battery field.

Response: We appreciate the reviewer's critical comment, which will strengthen our manuscript. Here we would like to provide deep insight on **two important findings (or breakthroughs) along with sufficient experimental, theoretical validations, and thorough comparisons with other technologies, similar approaches, and performances** laid out in this original manuscript, confidently saying, 'for the first time'; one for the *proposed methodology* and *physicochemical properties* of the resulting product (quasi-metallic silicon materials) and the other for *extraordinary electrochemical behaviors* compared with the previously reported bulk Si anodes.

① Scalable, cost-effective, and one-pot method

Figure R2. Required devices to perform the ion implantation (copyright. Justscience, 07.31.2017).

As shown in **Figure R2**, ion implantation method has been widely used in device engineering, particularly for forced injection of impurities to the various types of few inch-scaled wafers, which consists of more than five heavy and complicated equipments (ion source, ion extraction/ion separation magnet, ion accelerator, scanning system, and process chamber) to drive the ion sources. Although this process features several advantages over the previously applied 'diffusion process' such as precise control of doping level and relatively low temperature, high cost, intricate set-ups,

low injection depth of impurities (dopants), critical damages to the original substrates, and the potential hazard are not suited for the battery industry and hinder the further processing of impurity-embedded Si samples in the various energy applications. The estimated cost for the production of the specific substrates reaches about \$ 1,200 per a whole cycle (based on the cost announced in Korea Electrotechnology Research Institute), in which the size of the substrate is limited up to 6 or 10 inches, even though this process only passivates the top surface of the substrates. In addition, an interval between each running cycle of the instrument is too large to be fitted in large-scale production with poor processability.

Figure R3. Reduction reaction conditions and required devices to perform the low-temperature reduction method.

Apart from disadvantageous method described above, the proposed method here (referred to as 'low-temperature reduction method') enables a uniform doping of sulfur into the bulk but porous/hollow Si structure based on the co-growth mechanism of produced Si and S seeds from the low-cost precursors (importantly, the reduction products are independent of initial precursors) which features a facile, versatile, cost-effective, scalable, and one-pot approach. As shown in **Figure R3**, the reactor for this process just can contain about 3-4 g of precursors to produce an approximately 0.8-1 g of sulfur-doped Si (QMS) samples that is fairly close to the theoretical yields (40 % of silica) with a high purity of >95 at% Si (estimated cost <10 \$/kg_{Si} if the precursors are purchased in bulk-scale). In a practical manner, we can also construct a large-sized reactor which can accommodate hundreds of grams for precursors (or even more than thousands of grams for precursors). Otherwise, a bunch of reactors can be heated at the same time in the box furnace after sealed in the inert atmosphere but this kind of engineering on the reactor/systems would be out of scope for this work. We would rather want to focus on the practical feasibility of our proposed method for preparation of sulfur-doped Si samples in a powder form for the first time.

Figure R4. (a) Optical images of density comparison of two samples with a same amount (2 g). (b) Dissolution test of Li_2S in the electrolyte used here with a monitoring of color for 1 day.

Benefited from the co-growth mechanism of the seeds as evidently demonstrated in the manuscript and this revision, this simple method could insert the sulfur dopants as high as what it can be usually achieved from the ion implantation methods (usually 0.5-1 at%) without any additional forces. Concurrent seed growth process does not damage the entire structure as well as the crystallinity of typical cubic crystalline Si. Smaller atomic size of S seeds will be uniformly doped into the structure in over 3 μm scale. The unique structure itself is strongly favorable in the battery applications that require a high particle density of active materials. Compared with Si nanoparticles (< 0.1 g/cc), QMS particles have its density of higher than 0.4 g/cc as shown in **Figure R4a** (of course, making the shell of microparticles micro/macro-porous and coreless structure at once are also strong point). Without the conventional conformal carbon coating layers, QMS samples have a quasi-metallic property to show high electrical conductivity as experimentally confirmed in three different ways as well as theoretical calculations. This result is quite impressive because typical porous materials showed poor electrical conductivity compared with bulk crystalline materials due to enormous grain boundaries and contact resistance during measurement. If we consider its highly porous structure and large size, the genuine electrical conductivity will be potentially high enough to exceed that of other similar metals.

Figure R5. (a) Core-level XPS spectra of S 2p of hyperdoped Si samples, and for comparison, standard samples of pure S and (b) QMS (0.7) (a is reproduced from *Sci. Rep.* **5**, 11466 (2015)).

To our surprise as well, when recrystallization process occurs, S chains, not the sulfur atoms, could be directly inserted into the Si crystals to widen the interlayer spacings up to 0.5-0.7 nm compared with 0.31 nm of a lattice spacing for typical Si(111) plane. From the simulations, we found that the S chains inside the crystals are robust and flexible to sustain its structure even after completely cooled down to the ambient temperature. From XPS spectra, doping character of S into the Si is identical to the reported (**Figure R5**). This new finding was observed directly from the TEM analysis. Interestingly, internal channel created by the formation of such sulfur chains increased the lithium ion diffusion by 5,000 times from the GITT measurement, while the lithiation byproducts of lithium sulfide could help to maintain the diffusion path to a certain extent and metallic nature (Importantly, after the lithium sulfides are formed, it will not dissolve into the electrolyte as confirmed in **Figure R4b**).

② Stable and fast energy storage with high (initial) Coulombic efficiency

The prepared QMS here was evaluated as an anode material for LIB without any additional carbon or buffer layers which have been extensively studied in the literature. In general, Si materials could intake large amounts of lithium ions up to 3.75 Li⁺ per unit structure and this leads to large volume expansion/contraction during repeated charge-discharge cycles, which in turn raised critical issues such as particle pulverization, electrode delamination, and unstable SEI layers. Several strategies proposed to mitigate structural instability, such as nanostructuring, the unusual structure of yolk-shell/core-shell, the introduction of buffer layers, and multifunctional binders as described in the introduction of the original manuscript. However, most of them (predominant nano-scale designs)

are not suited for the practical purpose due to high cost, complicated preparation set-ups, and poor material density. Alternatively, there have been few attempts to utilize Si microparticles under the consideration of practical feasibility; coalescing the particles inside conformal graphene cage (*Nat. Energy* **1**, 15029 (2016)), self-healing binders (*Joule* **2**, 950 (2018)), highly elastic binders based on molecular pulley's action (*Science* **357**, 279 (2017)), and etc. Despite important breakthroughs from such designs, no one has achieved stable Si structure in micrometer scale without the buffering and supporting materials.

Remarkable electrochemical performances obtained from QMS anode can be attributed to three essential distinctions as follows:

- i) The resulting QMS itself has a fracture-resistant structure upon the huge stress generation because of abundant meso-/macropores and internal void spaces in spite of its bulk size (ca. 3 μ m) as primally demonstrated in our previous work (*Commun. Chem.* **1**, 42 (2018)). Interestingly, this unique structure even with uniform sulfur doping was enabled by one-pot low-temperature reduction reactions. As shown in the porous and hollow nano-Si (inner and outer diameter of 200 nm and 350 nm, respectively) anode (*Nat. Commun.* **6**, 8844 (2015)), we observed similar inward breathing of structure during lithiation in this bulk Si materials (QMS) and reversible accommodation of volume changes for hundreds of cycles. This implies that ensured structural stability of hollow/porous Si microparticles leads to interfacial stability as well.
- ii) Sulfur atoms doped into the substitutional positions in QMS significantly increased electrical conductivity by 50 times compared with that of undoped Si, which is related to the insulator-to-metal transition of typical chalcogen-doped Si. We carefully validated its conductivity (or conductance) in three different ways such as bulk conductivity based on the pelletized samples, single particle conductance from in situ measurement, and Hall effect measurement. Typical dopants of boron or phosphorous to Si materials will increase electric conductivity as high as what we can attain from QMS samples, but they require higher amounts of doping concentration (>3 at%) while available capacity of doped Si with B/P elements will be significantly lowered (*ACS Appl. Mater. Interfaces* **8**, 7125 (2016)). However, here with only 0.7 at% of sulfur fusion into the bulk Si materials, we could achieve good electric conductivity even higher than the carbon-coated Si. Also, minimized amounts of dopants increase available capacity during cycles. Importantly, the doping effect on Si anode was similar to other reports such as improved ICE (> 92%) and fast activation of particles (CE of 99.5% at 3rd cycle).
- iii) In terms of ion diffusion deep into the micrometer-sized structure, we observed sulfur chain-supported ion channels inside the Si structure. Such buffering chains are robust and flexible to sustain the original structure of Si without alteration on crystallinity and physical/chemical properties of Si. The sulfur chains will readily react with lithium ions to generate lithium sulfides nanoparticles (<2 nm) as demonstrated by both in-situ and ex-situ microscopy observations. From our theoretical calculations, we found that the lithium-sulfide-

embedded structure still sustained the fast diffusion path and QMS samples retained metallic character at the interface of amorphized Si and lithium sulfides, which help to enhance the lithium-ion diffusions as well as overall electrochemical properties during cycles.

Based on the above three major advantages of our approach, we render QMS anodes more stable and fast-chargeable without the need for carbon coating layers. (Please note that additional carbon coating on QMS will certainly enhance the electrochemical performances far beyond state-of-the-art battery anodes but this is not consistent with our report and will rather interrupt solid understanding of QMS samples first reported here. Additional buffer layers will restrict a clear interpretation of bare structure itself)

Specific comments:

Comment 2-1) *As some chalcogens including selenium or tellurium have been demonstrated to hold the ability of inducing insulator-to-metal transition of silicon, the sulfur doping of silicon in this work and its ability to induce insulator-to-metal transition is not originally and conceptually novel.*

Response 2-1: We appreciate the reviewer's critical comment which will strengthen our manuscript. As the reviewer is already noticed, the Si doped with chalcogen family including Se and Te along with S will show the insulator-to-metal transition (IMT) as we mentioned this point in the original manuscript at the beginning of the introduction and cited some important findings regarding Se/Te-doped Si wafers (ref. 44-46 and few more original literatures, 1) *Appl. Phys. Lett.* **38**, 9 (1981), 2) *J. Appl. Phys.* **51**, 7 (1980), 3) *Solid State Commun.* **226**, 1 (2016), etc.) via aforementioned complex ion-implantation method.

The bottom line of this work does not lie in the observation of insulator-to-metal transition phenomenon in the S-doped Si samples which have already been extensively studied with a special focus on its phenomenon, nature of carriers, and sub-band-gap optical properties. Instead, we report here a unique methodology to prepare the S-doped Si bulk powder samples in a scalable manner for the first time and their lithium storage properties evidently corroborated by theoretical calculations and cutting-edge experimental observations. In this original report, we thoroughly investigated what we obtained from the one-pot low-temperature reduction reactions (quasi-metallic bulk Si particles with hollow and porous structure) as a proof of concept, using the guidelines described in the above literature, thereby we tried to provide a general understanding of this new material to the potential readers in the material science field.

In the previous response, two strong points of our work are described in detail compared with conventional approaches and physicochemical properties of chalcogen-doped Si samples; 1)

Scalable, cost-effective, and one-pot method; 2) stable and fast energy storage with high (initial) Coulombic efficiency. Notably, we observed the diametrical lattice distortions (increase in lattice parameter evidenced by XRD patterns, TEM images, and simulations) in the Si crystals which originated from the buffered sulfur chains. The ion-implanted chalcogen-doped samples have never shown this trend in lattice parameters. They always demonstrated the contraction of the lattice in case of sulfur doping into the Si due to the solely substitutional doping character of the smaller size of S atoms than that of Si (*Phys. Chem. Chem. Phys.* **16**, 17499 (2014)). **We think that this unusual doping character driven by concurrent seed growth mechanism with the thorough evidence (along with the methodology itself) deserves to be considered conceptually and originally novel.**

Further, we left open the possibility of whether the other chalcogens (Se and Te)-doped Si samples will have similar properties or doping characters compared with what we demonstrated here to the potential readers in the conclusion section of the original manuscript. That is fair enough to arouse the reader's interest. However, we did not make any doubt of whether our proposed synthetic method will work for the Se and Te-involved systems because they all have similar properties regardless of their states of precursors and they all will be readily reduced in the molten salts medium to lead to the seed formation. Since Li *et al.* reported that Se-implanted Si shows minimum lattice distortion and formation energy at concentration of 1.56 % due to the ion-resonance interactions unlike the cases of S and Te doping (*Solid State Commun.* **226**, 1, (2016)), we are really excited about having a little different physicochemical property in case of Se doping in the near future through experimental validations which follow our proposed novel methodology. In addition, for the Te doping case, there is a strong possibility to obtain the air-/water-stable Si-Te compounds, not the Te-doped Si samples given that our doping methodology could induce the uniform distribution of dopants, although such compounds can be removed from the acid treatments (*Nano Lett.* **15**, 2285 (2015)). However, traditional ion-implantation could not achieve such unusual doping character and could not provide a proper subject matter of the study due to the highly destructive nature of method and alteration of the crystal structure of Si. Considering these experimental and theoretical backgrounds, sulfur doping into the bulk Si samples through our proposed novel methods is eminently suited for the model system to investigate new materials in multidirectional aspects.

By extension, we evaluated this new quasi-metallic bulk Si as a negative electrode material for LIBs from the fundamental measurements to the advanced in-situ characterizations. Despite the proof-of-concept, the QMS samples showed compelling electrochemical performances over the ever-reported Si microparticle materials. First of all, as demonstrated elsewhere, electrical conductivity of quasi-metallic state in QMS samples significantly increases the initial Coulombic efficiency (ICE) as high as 92.5% which is quite close to that of carbon-coated Si samples (*Commun. Chem.* **1**, 42, (2018)) without any buffer and coating layers. When introducing the carbon coating layers, we could not assume that such enhancements solely originate from the

conductivity itself or not, otherwise coming from the interfacial stability. However, infinitesimal sulfur fusion into the bulk Si structure obviously increases the electrical conductivity and consequently enhances ICE while maintaining the microstructure, morphology, and surface property. Further, sulfur chain-buffered ion channel inside the structure contributes to the enhanced initial reversibility as validated from the experimental and theoretical manners as well. Importantly, unique porous/hollow structure in the micrometer scale renders the QMS-based battery long-lasting without additional coating layers on it, which is supported by retained metallic nature of Li₂S-embedded amorphous Si and diffusion paths for lithium ions with low barrier energy. The newly added Supplementary Table 1 carefully compares the electrochemical performances of QMS anodes with other Si microparticle anodes either in primary or secondary structures, which indicates that our QMS anodes achieved superiority over the ever-reported anodes.

With all this in mind, we would like the reviewer to concede the novelty of this work not in the insulator-to-metal transition phenomenon itself but in the methodology, unusual doping characters, and thorough electrochemical validations along with in-situ/ex-situ characterizations and theoretical validations.

Comment 2-2) *With excessive structural discussion that could be moved to SI, the QMS remains unclear.*

Response 2-2: We appreciate the reviewer's critical comments which will strengthen our manuscript. We are not sure of fully grasping the intention of this comment, but we assume that the reviewer require in-depth and concise discussion on the structural evolution of QMS since it is the first time to develop the sulfur-doped Si samples not from the traditional ion-implantation method (Otherwise, please clarify the comment for us to have a chance to fully address the raised issue). The questions about the detailed description on how the new structure and doping mechanism can be developed unlike traditional method have been answered in the response to Reviewer #1 as attached below. Given that the methodology proposed in this article is completely different from what has been tried, we have to lay out a rather excessive but necessary discussion on the mechanism of structural evolution and characterizations.

We recognize that proposed doping mechanism in this original article might be less specific for the non-specialists (potential readers) in other fields as the reviewer pointed out. In fact, the proposed mechanism of sulfur dopants in the bulk Si materials is based on our previous report (*Commun. Chem.* **1**, 42 (2018)) that fundamentally demonstrated how the metal and metal salts form a complex structure, adsorption conformations and reduction activity of these species. Also, our 'solid experimental evidences' support such speculations. At above the melting point (T_m) of

metal salts (AlCl_3 , $T_m \sim 192\text{ }^\circ\text{C}$), which is significantly lower than the required thermal energy of typical thermochemical reduction process of silica (*e.g.*, Mg for $650\text{ }^\circ\text{C}$ and Al for $660\text{ }^\circ\text{C}$), these salts melt to form a molten salt medium inside a closed reactor and then solvate the Al metal at low temperature to make a Al- AlCl_3 complex. Afterward, activated AlCl^* from the ligand of the complex in two different thermodynamic pathways is spontaneously adsorbed onto the oxygen atoms of silica, thus producing a crucial byproduct of aluminum oxychloride (AlOCl) and creating the Si-Si/S-S/Si-S bonds. The successive reduction reactions provide a clustering environment of Si seeds and they are eventually recrystallized into spherical shell structure via localized Ostwald ripening process, while the byproducts and excessive molten salts contribute to macro-/meso-pores of hollow and porous Si. Importantly, the final morphology derived from this low-temperature reduction reaction does not depend on the morphologies/purity/chemical compositions of initial precursors (SiO_2 , clays, etc.), in which the same will apply in our S-doped Si systems and accordingly our approach features mass-scalability and versatility.

Evidently, we observed S-related products after the reduction reactions of sulfate salts precursor in two different formulations (one for sulfate reduction and the other for silica/sulfate reduction in presence of Al/ AlCl_3) as shown in **Figure R1**; XRD patterns confirm the formation of elemental sulfur reduced from the sulfate by Al- AlCl_3 . This result implies that high oxygen affinity of the complex (toward electron-rich elements such as oxygen, sulfur, etc.) will lead to a strong adhesion to O-edged surface of salt crystal (MgSO_4), whereby isolates (or creates) a sulfur seed along with formation of Mg-Al-Cl complex (MgAl_2Cl_8), and relatively small amounts of AlOCl as reported previously (*Energy Environ. Sci.* **8**, 3187 (2015)). Thus, during the reduction process in presence of both sulfate and silica, two different seeds will be formed and give a uniform mixture in the molten salt media, then finally sulfur seeds are doped into the Si host structure in micrometer scale unlike the surface-rich doping from the ion-implantation and destruction of the original structure.

In detail, we need to pay attention to formation of MgAl_2Cl_8 which was known as byproducts of the reaction in presence of metallic magnesium and AlCl_3 (*e.g.*, $2\text{Mg} + \text{SiO}_2 + 6\text{AlCl}_3 \rightarrow 2\text{MgAl}_2\text{Cl}_8 + 2\text{AlOCl} + \text{Si}$). This implies that, as we strongly assumed and even believed in this work, Al- AlCl_3 complex could selectively reduce MgSO_4 and then produces metallic Mg, amorphous sulfur (sulfur seeds) and AlOCl byproducts. Accordingly, the produced Mg will participate in subsequent reduction reaction as following the above equation and generate an unexpected product of MgAl_2Cl_8 . In other words, we revealed, for the first time, that high reactivity of the complex can reduce the ceramics (metal oxides) and even oxygen-containing salt crystals, thus demonstrating that our versatile approach can be extended to introduce the dopants in to the host structure from the seed level and yield uniformly doped materials according to specific purpose.

In order to clarify the reduction mechanism in this study, we added the following description and supplementary information to the revised manuscript as follows (Page 4, line 69-89);

“In principle, aluminum chloride (AlCl_3) used here as metal salts (T_m of 192°C) as well as molten salt medium can solvate bulk aluminum (Al) to form the highly reactive Al- AlCl_3 complex structure that spontaneously reduces different types of silica (e.g., cost-effective clay minerals and commercial bulk SiO_2) in two thermodynamically stable pathways. Activated AlCl^* from the ligand of the complex adsorbs on the oxygen atom of these compounds and then generates unusual byproducts of aluminum oxychloride (AlOCl) along with the formation of Si seed as demonstrated previously⁷. The complex species also react with an additional metal salt (MgSO_4), selectively dissociating the oxygen atoms from the salt crystal structure and thereby yield isolated magnesium (Mg) and sulfur. The presence of metallic Mg, which acts as the metal center of the complex, leads to the evolution of secondary byproducts of MgAl_2Cl_8 as clearly evidenced by X-ray diffraction (XRD) analysis of crude products after the reduction reactions (**Supplementary Fig. 1**). Further, protuberant XRD peaks observed at near 20° and 40° , correspond to the amorphous sulfur clusters while still buried with sharp peaks for either metallic Al or Si, which are indicative of the simultaneous formation of Si and sulfur seeds.

Strong reducing power of the complex completely disintegrate initial precursors into the active Si and sulfur at atomic level as suggested above but still stabilized by and embedded in the fluidic molten salt medium. Afterwards, abundant byproducts set the clustering environment for each atom to grow into the seeds that have a relatively free motion in the medium and eventually they are assembled into the spherical structure to reduce the surface energy of particles as the reactor cools down to ambient temperature.”

Supplementary Figure 1 | Seed formation analysis. a, b XRD patterns of crude products and samples after H₂O wash in (a) $\text{MgSO}_4\text{-Al-AlCl}_3$ and (b) $\text{SiO}_2\text{-MgSO}_4\text{-Al-AlCl}_3$ systems with other reaction parameters kept identical described in the experimental section.

Comment 2-3) *Although the increased electronic conductivity of the QMS is observed as shown in Figure 2, there is not any supportive evidence to correlate this increased electronic conductivity and the electrochemical performance, since the conversion of sulfur toward lithium sulfide upon cycling. That is, the section demonstrating electronic conductivity increase has little relationship with the main conclusion of the manuscript.*

Response 2-3: We appreciate the reviewer's critical comments which will strengthen our manuscript. As we explained the correlation of the increased electronic conductivity of the QMS and initial lithium storage property (ICE) and other related evidence regarding the above relationship in the **Comment 2-6**, the conductivity significantly impacted the initial reversibility of the battery without any doubt. However, the conductivity is not the only factor to determine the electrochemical performances. The sulfur chain-buffered ion channels obviously increased ion diffusions as verified by the GITT measurements, analysis of scan rate-dependent kinetics, and theoretical calculations, which also affected the initial electrochemical performances. The minimized capacity loss at the first cycle implies that the SEI layers were uniformly formed to suppress the further decomposition of electrolytes and promote the uniform Li-ion flux over the entire structure.

Apart from the improved initial performances of QMS electrodes, there might be concerns about how the conversion of sulfur into the lithium sulfide affects the cycling efficiency, cycle life, rate performances, and swelling property of QMS anode-based battery. Since we directly observed the formation of lithium sulfide in **Figure 4**, we set up the model system similar to the observed structure consisting of sub-nanometer-sized lithium sulfides embedded in the amorphized Si matrix (High ICE of QMS anode suggests almost the full activation (alloying reaction with Li-ion) of the particle and consequently crystalline structure of QMS will be transformed into the fully amorphized one) as shown in the **Supplementary Fig. 12**. *Li₂S provides an electron to the unoccupied conduction band minimum (CBM) state of internal amorphous Si, and QMS can have the metallic property by the charge transfer even after the QMS lose the original sulfur chain-buffered channels and crystallinity, unlike the other traditional dopants.* The charge density plots demonstrate that the CBM state without the lithium sulfides and metallic state with the lithium sulfides originate from the same internal Si. The other metallic band can be explained with the metallic surface states of the amorphous Si which has weaker metallic property since the number of states is not related with bulk volume, but the red CBM can be proportional to bulk volume. *Also, the Li-ion diffusion pathway through the interface between the lithium sulfide and amorphous Si has relatively lower barrier energy compared with that through the bulk structure of amorphous Si as corroborated by the retained high diffusion coefficient for Li-ion in the Supplementary Figure 11.*

As the reviewer pointed out, we also concerned about the sustainability of the improved electrochemical performances after the loss of sulfur dopants in the form of lithium sulfides at the beginning of this project. However, *we believe that the above observations evidently provide a decent correlation between the improved electrochemical performances with a little changed electrical conductivity but still metallic property as well as the fast diffusion pathways.* Of course, we are not trying to overstate the role of the sulfur and lithium sulfides, but the observations and various theoretical/experimental results just speak for themselves. The structural superiority over the other Si-based anodes also plays a crucial role in improving the overall specifications of the demonstrated battery. Also, we have already described the correlation between the conductivity and electrochemical performances after cycles in the original manuscript as follows (Page 8, Line 190-200):

“We assumed that the lithium-sulfide-related structure sustained the diffusion path, and our DFT calculations found a low diffusion barrier of 0.32 eV (Supplementary Fig. 12a,b). In our calculated model structure, the metallic property of QMS is demonstrated by the change in the CBM occupation (red band) owing to the charge transfer from the lithium sulfide to the amorphous Si. The unoccupied conduction state (red band) of amorphous Si (Supplementary Fig. 12c) is occupied by an electron from lithium sulfide, thereby maintaining its n-type character (Supplementary Fig. 12d). We assert that the metallic property of the QMS containing lithium sulfide nanoparticles mostly arises from the occupied CBM states of amorphous Si via charge transfer from the lithium sulfide. The charge density plots of the CBM state of H-passivated amorphous Si show features similar to those of the metallic state in the lithium sulfide–Si model system, demonstrating fast charge transfer.”

Supplementary Figure 12. Formation of lithium sulfide inside QMS. **a** Schematic illustration of Li diffusion path at an interface between lithium sulfide (Li_2S) and amorphous Si (a-Si). **b** Calculated diffusion energy barrier for Li-ion via the interface path. **c** Calculated band structure of the interface structure of H-passivated amorphous Si without Li_2S (Fermi level (E_f) set to zero) and charge density plot at CBM state (red line). White atoms on silicon surface are hydrogen atoms. **d** Calculated band structure of the interface structure of a-Si with Li_2S and charge density plot at the state across the E_f (red line).

Comment 2-4) *The authors overstated the role of sulfur or lithium sulfide in electronic and ionic conductivity; the comparison and discussion were excessively focused on the role of sulfur or lithium sulfide. In the referee's opinion, other factors, for example, the pore structure, may mainly contribute to the observed difference. Yet, given the pore structure modulation, the novelty of the work is limited again while pore engineering has been widely demonstrated.*

Response 2-4: We appreciate the reviewer's critical comments which will strengthen our manuscript. We think that the reviewer assumes that major differences in the electrochemical measurements do not mainly arise from the sulfur doping-enabled physicochemical properties, but from the porous structure of the samples which have been extensively studied in the literature. It should be noted that *the undoped and doped Si samples have the same hollow and porous structure with almost the same particle size distribution and adsorption properties* as shown in the **Supplementary Figures 3-5 and Figure R6**. That is why we focused on the sulfur-doping-driven changes in the structural and electrical properties and their correlation with electrochemical properties throughout the manuscript. *Understanding of how infinitesimal amounts of sulfur doping could make such stark differences is the bottom line of this work and also the methodology to enable the unusual doping characters deserves to be considered novel enough to arouse great attention of the potential readers.*

Figure R6. TEM images of (a) undoped Si and (b) QMS(0.7) samples.

In our previous report (*Commun. Chem.* **1**, 42 (2018)), we revealed the unique mechanism of molten-salt induced low-temperature reduction systems and resulting products (referred to as hollow and porous Si sphere, HPSS) are identical to the undoped Si in this work. Previously, we applied a traditional strategy of introducing the amorphous carbon layers via chemical vapor deposition process to further enhance the electrochemical performances. The superior volume-accommodating ability of hollow and porous structure was already validated by in-situ/ex-situ TEM analysis and relevant electrochemical tests. That is, the effect of engineered pores in the QMS samples has been thoroughly compared with other reports in terms of their electrochemical figure of merits.

Thus, we conducted step-by-step characterizations on the QMS itself regarding the property changes originated from the sulfur doping. First significant alteration is the insulator-to-metal

transition which was validated by three different methods, thereby confirming the improved electric conductivity of QMS samples which greatly impacted the initial lithium storage properties as discussed in the **comment 2-6 and shown in Figures 1 and 2**. Second unusual changes occurred inside the Si crystal structure. The S seeds are incorporated between the Si(111) planes by chance during the recrystallization process which was directly observed from the TEM analysis. Importantly, we found that the buffering sulfur chains are robust and flexible to widen the interior spaces which accelerate internal diffusion of Li-ion. Although the lithium ions could diffuse inside such channels without any barrier or activation energies, it will eventually lead to the formation of lithium sulfide inside the Si structure as shown in **Figure 3**. Then, we carefully investigated the correlation of amorphized Si structure embedded with lithium sulfide after the lithiation with its electrochemical behavior (**Figure 3-5**). In **Figure 4g**, the size of lithium sulfide embedded in the amorphous Si was approximately 1-2 nm. Based on this observation, we set up the model structure similar to the observation and investigated the interfacial property as shown in the **Supplementary Figure 12**. Compared with the lithium sulfide-absent systems, amorphous Si with the lithium sulfide showed a metallic property due to the charge transfer at the interfaces. In addition, we found the diffusion pathway with lower barrier energy compared with that through the bulk structure of amorphous Si. The retained metallic nature and diffusion pathway with low barrier energy facilitate the fast diffusion behavior of Li-ion during subsequent cycles as shown in the **Supplementary Figure 11**. *Benefited from these advantageous physicochemical properties of QMS samples before and after electrochemical cycling, not solely from the porous structure, the QMS electrodes have compelling performance both in the half and full cell.*

Comment 2-5) *The QMS shows relatively improved performance compared to its counterpart provided in this work. Yet, the QMS performance either in half cells and full cells is rather mediocre, and the significant capacity decay was still observed upon cycling.*

Response 2-5: We appreciate the reviewer's critical comments which will strengthen our manuscript. We accept that the reviewer might think so if this work is fragmentally compared with other works previously reported. In fact, capacity fading observed in **Figure 5b** was not serious but rather impressive; after 500 cycles, the QMS(0.7) electrodes still delivered the areal capacity of 2.58 mA h cm⁻², which is achieved without carbon coating layers or Li metal change in other literature (*Science* **357**, 279 (2017)). Therefore, we prepare the comparison table in the revised supplementary information (**Supplementary Table 1**) as shown in the below (half-cell results) and additional plot for volumetric/gravimetric energy density of other promising full cells to compare the full cell performances only displayed in the revision (**Figure R7**).

The summarized recent progress of micron-sized Si anodes mainly includes two types of structure. Firstly, the direct use of Si microparticles (SiMPs) as an anode material was demonstrated in a few works with various types of carbon coating layers. Li *et al.* introduced conformal graphene cages on the SiMP but its coating process was immensely complicated. Also, the capacity retention was not as good as that of QMS anodes also with limited rate performances in spite of high ICE of 93.2%. Except for this outstanding comparison sample, other coating strategies recently published or published in high impact journal did not meet the overall electrochemical performances compared with compelling performances of QMS anodes without any additional carbon coating layers.

Second predominant strategies are based on constructing the secondary Si/C structure with Si nanoparticle (SiNP) building blocks and carbon buffers. These secondary structures neither achieved ICE values higher than 92% nor high areal capacity loading ($> 3.5 \text{ mAh cm}^{-2}$) although the carbon buffers and SiNPs are rather stable during hundreds of cycles but still far behind what we achieved from the QMS anodes here. Although these secondary structures using nanoscale building blocks are totally different from our design concept of bulk structure and have been considered as an ideal structure for the future applications, the proposed here of QMS anodes are much suited for high-energy-density LIBs based on the practical considerations (*Nano Res.* **10**, 3970, (2017)).

Apart from the electrochemical evaluation in the infinite lithium sources (counter and reference electrodes are Li metal), full cell evaluation of the prepared electrodes becomes much crucial to consider the practical feasibility. We take four other representative high-energy-density LIBs based on the Si-based anodes and the traditional or high-voltage LCO cathodes, which was reported in high-impact journals with an emphasis on their improved energy densities (**Figure R7**). We applied the same calculation methods for the volumetric and gravimetric energy density of the full cells (Otherwise, we reuse the figures described in the published article). Notably, our QMS/LCO full cell delivered the highest both volumetric and gravimetric energy densities compared with other promising designs based on the Si-based anodes and similar LCO cathodes.

With these rigorous comparisons in terms of its performance matrixes, we would like to rather highlight our significant figure of merits over other featured designs of Si-based anode unlike the concerns from the reviewer. Accordingly, we revise the manuscript to highlight the achieved high energy density of the constructed full cells as follows (Page 9, Line 230-234):

*“The decreased polarization from the metallic nature of QMS and the partially retained ionic channels during cycling lead to a **high volumetric/gravimetric energy density of full cells compared with other promising designs for Si-based anodes (Supplementary Table 1 and Supplementary Note 2)**”*

Supplementary table 1. Comparison chart for recent progress of micron-sized Si anodes

Materials	Primary and/or Secondary particle size (μm)	Electrode composition (A:B:C)	ICE (%)	Areal capacity loading (mA h cm^{-2})	Initial reversible capacity (mA h g^{-1})	X, Capacity retention (%) after Y, cycles at Z, current density (A/g)			X, Capacity (mA h g^{-1}) and Y, capacity retention (%) ^e at Z, Max. discharge current density (A/g)			Ref.
						X	Y	Z	X	Y	Z	
QMS(0.7)	ca. 3	80:10:10	92.5	~3.8	3,350	72	500	1.75	1,100	>35	17.5	This work
Si		80:10:10	87.4	~3.3	3,080	<10	500	1.75	0	0	17.5	7 ^f
HPSS@C	ca. 3	80:10:10	91	~3.4	3,494	57	800	3.5	~1000	~33	17.5	7 ^f
Bulk porous Si@C	1.5 N/A ^a	70:0:30	92	3.4-4.5	~2,100	92.4	50	0.2	<1500	~50	2	4
mSi@OG ^b @RGO ^c	1-5 N/A	80:10:10	78	~2.8	~2,500	~90	150	2.0	<500	<20	12	5
p-Si/C	<0.2 ca. 5.28	70:20:10	~75	2.8	~2,500	83	370	2.6	~700	~20	11	6
SiMP@SiC/a-SiO _x /C/a-Li ₂ SiO ₃	~1 N/A	70:10:20	77.7	~2.3	1,924	~75	100	0.1	1,439	N/A	0.1	7
Si/C	<0.1 ca.14.8	90:5:5	89.2	~2.85	620	0.75	500	0.06	~500	80	3.0	8
Si-MCS	<0.1 ca. 2	60:20:20	75	~2.44	~1,350	94	500	0.8	880	73	40	9
Si@Gr	1-3 N/A	90:10:0	93.2	~3.0	~3,300	~75	325	2.1	~350	~10	16.8	10
FeCuSi	<0.15 ca. 6.5	80:10:10 (Si) 97:1.5:1.5 ^d	91 91.4	~1.3 3.44-3.48	1,287 420	~60	300	0.42	~252	N/A	0.42	11

Note: ^aNot applicable. ^bOverlapped graphene. ^cReduced graphene oxide. ^dGraphite-blending electrode. ^eCapacity retention at max discharge current density compared with that measured at initial current density during rate capability test. ^fReference in manuscript.

Figure R7. Comparison of volumetric/gravimetric energy density of the full cell with QMS anodes and recent achievements. The calculations and detailed information are referred to *Energy Environ. Sci.* **8**, 2075 (2015), *Adv. Energy Mater.* **4**, 1301542 (2014), *Nat. Energy* **1**, 16113 (2016), and *Nat. Commun.* **8**, 812 (2017) displayed in order.

Comment 2-6) The authors show improved ICE, yet the related explanation is missing. In fact, it is very strange for the referee to see this improvement without any interface modification.

Response 2-6: We appreciate the reviewer's critical comments which will strengthen our manuscript. As the reviewer pointed out, one of the important points in this work lies in the achieved high ICE of 92.5% which is quite high if we consider the high porosity of materials, thus requiring proper rationales based on the unique structure and physicochemical properties. For a deep insight on the improved ICE values of QMS samples, we would like to lay out different aspects to determine the initial reversibility from the historical viewpoints going through our previous work and some critical papers to regulate the ICE of the Si-based anodes.

<Porosity and Conductivity>

In the general understanding of the correlation between the porosity and ICE, there have been a lot of reports that showed the lower ICE of the highly porous electroactive materials including Si, Ge, and Sn while effectively accommodating the large volume changes of such high-volume-change materials. Liu and Zhang *et al.* reported highly mesoporous Si sponge anode with a bulk particle size of 20 μm and dominant 10-50 nm size of pores (*Nat. Commun.* **5**, 4105 (2014)). The ICE of this sample was only ~56% due to the abundant mesopores and the high oxygen content in the surface of samples, although typical SiMPs with a high crystallinity have a relatively high ICE values (*Joule* **2**, 950 (2018)).

→ Micro/Mesopores should be avoided

In addition, Park and Cho *et al.* prepared 3D microporous SiMPs with carbon coating layers of which ICE reached 94.4% compared with that of non-coated Si samples (69.3%, *Adv. Energy Mater.* **2**, 878 (2012)). The authors described that the remarkable ICE may be attributed to the interfacial engineering, but we think that the conductivity will have much significant effect on the ICE. Accordingly, in 2017, Song and Seo *et al.* carefully compared the correlation between the porosity and conductivity based on the doped Si NWs with and without the porosity, labelled as *l-m/h*-SiNW (*J. Electrochem. Soc.* **164**, A1564 (2017)). As shown in table of **Figure R7**, the increased electrical conductivity of *m*-SiNW leads to a great enhancement in ICE from 51.5 % of *l*-SiNW to 88.5 % when its surface area maintained as low as 16 $\text{m}^2 \text{g}^{-1}$. If the conductivity factor ensures high initial reversibility, then, additional pore generations on the SiNW (increased surface area by 16 times) did not lead to large irreversible capacity loss. Interestingly, *h*-SiNW with high conductivity but high surface area results in rather higher ICE than other electrodes.

→ Priority: Electric conductivity + Electrolyte accessibility > Electric conductivity > Surface area

Figure R7. Physicochemical properties of SiNWs and Si-Li alloying reaction in a coin cell configuration of SiNW in a carbonate-based electrolyte. Loading density of Si = 0.6 mg cm⁻². (a) Potential profiles at the initial galvanostatic lithiation and delithiation at 0.05 C and (b) Capacity retention of SiNWs during repeated cycles of lithiation and delithiation at 0.2 C. Reproduced from *J. Electrochem. Soc.* **164**, A1564 (2017).

In the meantime, Park *et al.* reported the multiscale hyperporous silicon flake anodes which achieved high ICE of 92.5% in spite of high surface area of 188 m² g⁻¹. This unique structure involved dominant macropores occupying over 80% of total pore volume while minimizing the mesopores and micropores that do not significantly contribute to accommodation of large volume change but consumes large amounts of electrolyte to give a lower ICE. Also, they added “the electrolyte can be easily infiltrated into the highly accessible structure (open framework) to activate quickly overall electrode components”. In this porous structure, carbon coating layers which accelerate electron transfer and enable the stable interfaces in general have no impact on the ICE (92.6% → 92.7%).

→ **Desirable macropores or open framework for quick activation of particles and in such case, additional conductivity contribution is not critical**

All things considered, conductivity factors should be considered first and mesopores/micropores should be avoided. Building the Si structure into the open framework is not necessary but favorable to achieve high ICE. In this context, **the metallic nature of QMS samples investigated in this work critically contributes the high ICE along with three other beneficial features.** First, highly accessible hollow and porous structure of QMS leads to fast activations of the overall electrode. Also, the frames of QMS samples are connected each other to further facilitates electron transfer and Li-ion diffusion. Second, dominant macropores and minor portion of meso/micropores of QMS suppresses unnecessary consumption of electrolytes. Lastly, formation of internal ion channel for Li-ion greatly enhances the diffusion kinetics. Benefited from the above four main factors, QMS electrode without any additional carbon coating layers could achieve the high ICE. In order to clarify this point, we revised the manuscript as follows (Page 8-9, Line 211-218):

“As systematically investigated in the literature⁴², the conductivity factor much more critically determines the initial reversibility of Li-ion transport in the Si-based anodes with other

physicochemical properties being equal than the factor from the surface area of the samples. Further, the highly accessible hollow and porous structures for the electrolytes facilitate quick activation of the entire electrode and the sulfur-buffered Li-ion diffusion channels enhance the diffusion kinetics. The dominant macropores and a minor portion of meso/micropores are also beneficial to achieve the unprecedented high ICE of 92.5% without additional carbon coating layers.”

Comment 2-7) *Overall, the referee has concerns about its novelty, importance and the impact on this field. Thus, it is hard for me to recommend publication of this manuscript in Nature Communications.*

Response 2-7: With the above point-by-point responses that hopefully but fully address those concerns raised by the reviewer, we suggest the reviewer to reconsider the recommendation of our work to the publication in *Nature communications*. Here are the main achievements (or summary) of this original work which hopefully help the reviewer to understand novelty of our work:

i. New Methodology

Compared with the traditional method of ion-implantation and followed the intricate process, the proposed methodology (referred to as ‘low-temperature reduction reactions’ to facilitate concurrent seed growth mechanism) enables the uniform sulfur doping over the Si microparticles. The strong reducing agents of Al-AlCl₃ complex fragment all the precursors including silica and sulfate to create atomic-scale Si and S. They will be readily clustered in the molten salt medium and then recrystallized into the hollow/porous structure. This process, *for the first time*, can produce the powder form of sulfur-doped Si (QMS) samples in the scalable and practical manner. From the various physicochemical measurements, the QMS samples showed high electric conductivity (increased by 50 times) which is similar to that produced from the traditional method while its doping depth and uniformity are far beyond that achieved from the traditional method.

ii. New doping characters

The developed QMS have a different doping character compared with the substitutional doping of chalcogens as well as typical dopants (B, N, P). During the clustering process of the S and Si seeds, the *chain-like structure of S can be directly inserted into the cubic crystal Si*, which prevents the bond saturation of Si planes and rather widens the lattice parameters contrary to the conventional understanding. The TEM observations are consistent with our

theoretical calculations that demonstrated the possible incorporation of sulfur chains inside the Si crystals with its high flexibility and robustness. This could provide the diffusion channel for Li-ion without requiring any barrier energy to diffuse deep into the structure that significantly increases the diffusion coefficient by 5000 times in the GITT measurement along with the theoretical validations.

iii. Sulfur-embedded structures

After the formation cycle of the battery, as expected and well-known, sub-nanometer-sized lithium sulfides are embedded in the Si structure which will not be further agglomerated in the following cycles. Thus, we focused on the theoretical demonstration of this new structure regarding the diffusion of Li-ion and electrical conductivity. Interestingly, we found the fast diffusion pathway through the interfaces between the amorphous Si and lithium sulfides with low barrier energy for the diffusion and retained metallic property at the interface. This closely correlates with improved rate performances and improved long-term stability of battery.

iv. Remarkable battery performances

As a proof of concept, we constructed the half and full cell with the QMS electrode. At the first cycle, QMS electrode has a high ICE of 92.5% compared with 87.4% of undoped Si in addition to the increased available capacity of QMS electrode due to the deep diffusion of Li-ion through the ion channels, increased conductivity, and highly accessible structure for Li-ion in QMS electrode. During the subsequent cycles, the retained diffusion pathway and high electric conductivity lead to stable cycle retention of 72% after 500 cycles at 1.9 mA cm^{-2} at extremely high areal capacity loading of 3.8 mA h cm^{-2} as well as excellent rate capability despite its bulk structure. Importantly, we found that the QMS electrode showed improved electrode swelling result as low as 50% in spite of that the QMS electrode consisted of pure Si without any buffer layers. Also, the assembled full cell of QMS/LCO showed the highest gravimetric/volumetric energy density over the state-of-the-art Si-based anode/LCO combinations. This suggests that the QMS electrode can be considered as practically viable and feasible systems to advance the Si-based high-energy-density LIBs in the near future.

REVIEWERS' COMMENTS:

Reviewer #1 (Remarks to the Author):

The scientific contents have been much improved after revision. The reviewers' comments have been fully addressed with many additional experimental data and relevant scientific explanation. The reviewer believe that revised manuscript will attract broad scientific and industrial interests in the field of energy storage via lithium-ion batteries. Therefore, I would like to recommend the acceptance of the revised manuscript for publication.

Reviewer #2 (Remarks to the Author):

The paper has been revised substantially to a level publishable in Nature Communications as it is.

Soojin Park, Ph.D.

Professor

Department of Chemistry, Pohang University of Science and Technology

77 Cheongam-Ro, Pohang 37673, Republic of Korea

E-mail: Soojin.park@postech.ac.kr

Phone: 82-54-279-2102

Homepage: <https://www.spark-postech.com/>

April 25, 2019

Manuscript ID: NCOMMS-19-00539-A

Responses to all editor/reviewer comments for Manuscript ID: NCOMMS-19-00539-A

(**Title:** Infinitesimal sulfur fusion yields quasi-metallic bulk silicon for stable and fast energy storage)

Reviewers' comments as follows;

Reviewer# 1

Comments:

The scientific contents have been much improved after revision. The reviewers' comments have been fully addressed with many additional experimental data and relevant scientific explanation. The reviewer believe that revised manuscript will attract broad scientific and industrial interests in the field of energy storage via lithium-ion batteries. Therefore, I would like to recommend the acceptance of the revised manuscript for publication.

Response: We thank the reviewer for the critical and comprehensive understanding of our work with constructive comments, whereby we hope that our findings, systematic methodology, and remarkably improved performances will draw a great attention from materials science society as the reviewer suggested.

Reviewer# 2

Comments:

The paper has been revised substantially to a level publishable in Nature Communications as it is.

Response: We thank the reviewer for the critical and comprehensive understanding of our work with constructive comments, whereby we hope that our findings, systematic methodology, and remarkably improved performances will draw a great attention from materials science society as the reviewer suggested.